# A Spectral Approach to Generalization and Optimization in Neural Networks

## Abstract

The recent success of deep neural networks stems from their ability to generalize well on real data; however, Zhang et al. (Zhang et al., 2016) have observed that neural networks can easily overfit random labels. This observation demonstrates that with the existing theory, we cannot adequately explain why gradient methods can find generalizable solutions for neural networks. In this work, we use a Fourier-based approach to study the generalization properties of gradient-based methods over 2-layer neural networks with sinusoidal activation functions. We prove that if the underlying distribution of data has nice spectral properties such as bandlimitedness, then the gradient descent method will converge to generalizable local minima. We also establish a Fourier-based generalization bound for bandlimited spaces, which generalizes to other activation functions. Our generalization bound motivates a grouped version of path norms for measuring the complexity of 2-layer neural networks with ReLU activation functions. We demonstrate numerically that regularization of this group path norm results in neural network solutions that can fit true labels without losing test accuracy while not overfitting random labels.

## 1 Introduction

Deep neural networks (DNNs) have achieved state-of-the-art performance on a wide array of diverse tasks (LeCun et al., 2015). A given DNN architecture represents a highly rich space of hypotheses. However, numerous empirical results have demonstrated that a simple stochastic gradient descent (SGD) learner can efficiently search over this space to find a solution that achieves high performance on both training and test data. Despite many successful applications of DNNs to practical tasks such as computer vision (Krizhevsky et al., 2012), natural language processing (Collobert & Weston, 2008) and speech recognition (Hinton et al., 2012), our basic understanding of the factors that drive DNN generalization is still lacking.

Addressing generalization for DNNs is hard for two fundamental reasons: 1) Empirical risk minimization for neural networks is a non-convex optimization problem with possibly many local minima, and 2) Two different local minima with the same training performance can achieve significantly different performance on test data. For these reasons, the neural network optimization method plays an important role in the generalizability of the local minima found. For example, SGD has been empirically shown to outperform large-batch gradient descent (Keskar et al., 2016). Also, the performance of gradient methods can be improved upon by incorporating the geometry of observed data (Duchi et al., 2011; Neyshabur et al., 2015a).

For DNNs, however, a good optimization method is not sufficient for guaranteeing good generalization. Zhang et al. (Zhang et al., 2016) empirically demonstrate that a neural network trained by SGD can easily overfit random labels on the CIFAR-10 (Krizhevsky & Hinton, 2009) data. Yet, the same neural network fitted by the same SGD algorithm achieves good generalization performance for the original CIFAR-10 labels. This observation challenges the ability of traditional learning theory to explain why SGD learns generalizable hypotheses over neural networks. To shed light on this phenomenon, two recent works have developed generalization bounds and complexity measures for neural networks which can distinguish the local minima found for true and random labels. (Bartlett et al., 2017) proves a margin-based generalization bound and shows how it correlates with the generalization risk of DNNs when fitting true and random labels. (Neyshabur et al., 2017) explores different complexity scores for DNNs and how they behave differently for true and random labels. The complexity measures investigated in these works can effectively distinguish generalizable from poorly-generalizable local

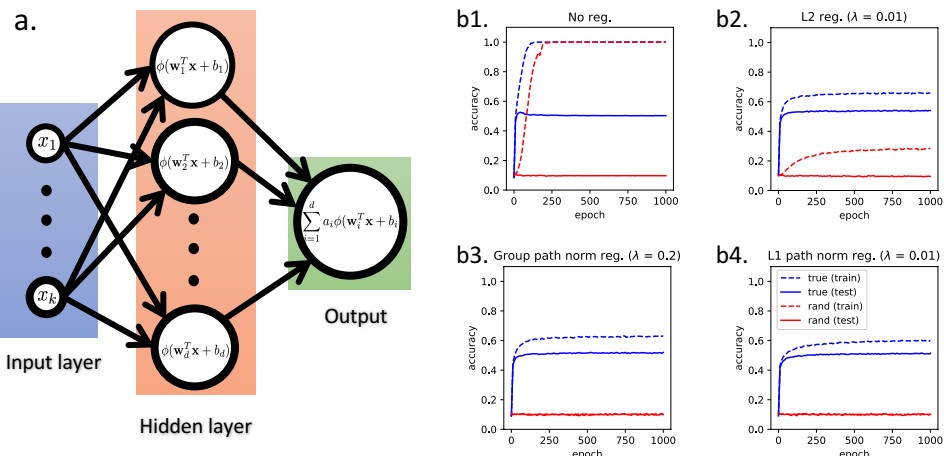

Figure 1: (a) A 2-layer neural network with activation function $\phi$, (b) Training and test accuracy on CIFAR10 with true and random labels on a 2-layer neural network with 512 ReLU hidden units, regularized with an additive penalty: (b1) no penalty, (b2) $\ell_2$-norm, (b3) $\chi_2$-group path norm, (b4) $\ell_1$-path norm. The $\chi_2$-group path norm and $\ell_1$-path norm were successful to close the generalization gap for both true and random labels.

minima. They do not explain, however, why SGD converges to generalizable local minima when there exist poorly-generalizable local minima which can also perfectly fit the training set.

To approach this question, one needs to understand the key characteristic of CIFAR-10's original labeling which differentiates it from random labels and how it is exploited by SGD to achieve good generalization performance. In this work, we approach this problem in the Fourier domain where non-random labeling schemes behave completely differently from random labeling schemes. While signals recoverable from few measurements possess nice spectral properties such as bandlimitedness, fully random stochastic processes are not bandlimited and not recoverable from any finite number of measurements.

Using spectral analysis, we focus on characterizing spectral properties of an underlying distribution which can be exploited by gradient-based methods to converge to generalizable local minima. We address this problem for 2-layer neural networks (see Figure 1a) with sinusoidal activation functions, where we show that if the underlying labeling scheme has limited bandwidth and Fourier $\ell_1$-norm (i.e. "nice" Fourier properties), we expect a gradient-based method to achieve good generalization performance. To arrive at this result, we first develop a Fourier-based generalization bound for 2-layer neural networks in terms of bandwidth and Fourier $\ell_1$-norm. Next, we prove that the local minima found by the gradient descent method over a 2-layer neural network with sine activation have bandwidth and Fourier $\ell_1$-norm bounded in terms of the spectral properties of the underlying labeling scheme.

As a byproduct of our Fourier analysis, we derive generalization bounds for 2-layer neural networks with general activation functions. For bandlimited activation functions with finite Fourier $\ell_1$-norm, such as sinusoidal or Gaussian activation[1], our bound is tighter than the generalization bound obtained using only the Lipschitz constant of the activation function. For ReLU-type activation functions, our generalization bound is comparable to Lipschitz-based bounds; however, it leads to a grouped version of the path norms developed in (Neyshabur et al., 2015a). We therefore call this capacity norm *group path norm* which can be used as an additive penalty to regularize 2-layer neural networks with ReLU activation. Our numerical experiments suggest that the generalization gap can be effectively tightened by regularizing the group path norm. Figure 1b demonstrates how group path norm regularization can help close the generalization gap for both true and random labels.

---

[1]Since a Gaussian function's Fourier transform can be arbitrarily well approximated by bandlimited functions in the $\ell_1$-norm sense, the results shown in this work are applicable to Gaussian activation.

## 2 RELATED WORK

Generalization has been a topic of central interest in statistical learning theory (Vapnik, 1999; Shalev-Shwartz & Ben-David, 2014). Generalization bounds have been derived using the stability of a learning algorithm (Bousquet & Elisseeff, 2002) and various complexity measures of a function space such as VC-dimension (Vapnik, 2013) and Rademacher complexity (Bartlett & Mendelson, 2002). (Hardt et al., 2015) develops a stability-based generalization result for SGD as the learning algorithm, which holds for both convex and non-convex loss functions.

We note that spectral analysis has provided a powerful framework for studying neural networks. (Barron, 1993) uses a Fourier-based approach to prove the universal approximation theorem for 2-layer neural networks. Similarly, (Lee et al., 2017) applies Fourier analysis to extend Barron's result to a general feedforward neural network. (Rippel et al., 2015) uses a spectral approach to model and analyze convolutional neural networks (CNNs) and introduce the spectral pooling scheme for CNNs. Also, our Fourier-based approach to analyze SGD's performance for 2-layer neural networks follows the same prinicples as the analysis performed in (Shamir, 2016) to prove the hardness of fitting periodic labeling schemes via gradient-based methods. We should note that in this work we use only periodic activation functions and not periodic labeling schemes. Therefore, the hardness result shown in (Shamir, 2016) does not affect our numerical experiments.

In general, theoretical studies of neural networks can be categorized into three main categories: 1) Approximation: Neural networks have been proven to be powerful in expressing a very rich class of functions (Cybenko, 1989) and in general deeper networks need fewer neurons to express the same class of functions (Eldan & Shamir, 2016; Liang & Srikant, 2016). 2) Generalization: Tight bounds have been shown on the VC dimesnion of feedforward neural networks (Anthony & Bartlett, 2009; Harvey et al., 2017). Also, norm-based Rademacher complexity bounds have been developed at (Bartlett & Mendelson, 2002; Neyshabur et al., 2015b). Sharpness of local minima and its connection to their generalizibility have been the focus of several recent works (Keskar et al., 2016; Dinh et al., 2017; Neyshabur et al., 2017) 3) Optimization: theoretical studies have shown both positive (Andoni et al., 2014; Daniely, 2017) and negative (Shalev-Shwartz et al., 2017) results about the performance of gradient-based methods in training neural networks.

## 3 PRELIMINARIES

### 3.1 SUPERVISED LEARNING AND GENERALIZATION

Suppose that we are given $n$ samples $(\mathbf{x}_i, y_i)_{i=1}^n$ drawn i.i.d. from a population distribution $P_{\mathbf{X},Y}$. Here $\mathbf{X}$ denotes the random vector of features and $Y$ denotes the target variable. Using these $n$ samples, the goal of a supervised learner is to find a prediction rule $f$ from a function space $\mathcal{F}$ which can predict $Y$ for an unseen test sample $\mathbf{X}$. Therefore, given loss function $\ell$ the supervised learner wants to find $f^* \in \mathcal{F}$ minimizing the population risk, defined as $\mathbb{E}\big[\ell\big(f(\mathbf{X}), Y\big)\big]$ averaged under the population distribution.

However, the supervised learner does not know the population distribution $P_{\mathbf{X},Y}$ and has only access to the $n$ training samples. The supervised learner can minimize the empirical risk, defined as $1/n \sum_{i=1}^n \ell\big(f(\mathbf{x}_i), y_i\big)$ and find $f_n^{\text{emp}}$. Since we only observe a limited number of samples, the empirical risk would be different from the population risk. The generalization risk, defined for $f \in \mathcal{F}$ as $\mathbb{E}[\ell(f(\mathbf{X}), Y)] - \frac{1}{n} \sum_{i=1}^n \ell(f(\mathbf{x}_i), y_i)$, is the difference among the population risk and empirical risk for $f$. Studying the behavior of $f_n^{\text{emp}}$'s generalization risk for different function spaces and learning algorithms is a topic of central interest in statistical learning theory.

### 3.2 FOURIER TRANSFORM AND BANDLIMITED FUNCTIONS

Consider a real-valued function $f : \mathbb{R}^k \to \mathbb{R}$. The Fourier transform of this function, which we denote by $\widehat{f}$, is defined as

$$\widehat{f}(\boldsymbol{\xi}) = \int f(\mathbf{x}) \exp\left(-2\pi i \boldsymbol{\xi}^T \mathbf{x}\right) \mathrm{d}\mathbf{x}. \tag{1}$$

Some important examples of Fourier transform are:

- *Sinusoidal function:* $f(\mathbf{x}) = \exp(2\pi i \boldsymbol{\omega}^T \mathbf{x})$, then $\widehat{f}(\boldsymbol{\xi}) = \boldsymbol{\delta}(\boldsymbol{\xi} - \boldsymbol{\omega})$ where $\boldsymbol{\delta}$ denotes the Dirac delta function, which also implies
  - $f(\mathbf{x}) = \cos(2\pi \boldsymbol{\omega}^T \mathbf{x})$, then $\widehat{f}(\boldsymbol{\xi}) = 1/2 \left[ \boldsymbol{\delta}(\boldsymbol{\xi} + \boldsymbol{\omega}) + \boldsymbol{\delta}(\boldsymbol{\xi} - \boldsymbol{\omega}) \right]$.
  - $f(\mathbf{x}) = \sin(2\pi \boldsymbol{\omega}^T \mathbf{x})$, then $\widehat{f}(\boldsymbol{\xi}) = i/2 \left[ \boldsymbol{\delta}(\boldsymbol{\xi} + \boldsymbol{\omega}) - \boldsymbol{\delta}(\boldsymbol{\xi} - \boldsymbol{\omega}) \right]$.
- *Gaussian function:* $f(\mathbf{x}) = (\sqrt{2\pi}\sigma)^k \exp\left(-\|\mathbf{x}\|_2^2/2\sigma^2\right)$, then $\widehat{f}(\boldsymbol{\xi}) = \exp\left(-\sigma^2\|\boldsymbol{\xi}\|_2^2/2\right)$. Thus, the Fourier transform of a Gaussian function preserves the Gaussian shape.

A function $f$ is called *B-bandlimited* if $\widehat{f}(\boldsymbol{\xi}) = 0$ for every $\boldsymbol{\xi}$ where $\|\boldsymbol{\xi}\|_2 > B$. The smallest $B$ for which this property holds is called the *bandwidth* of $f$. We use $\mathcal{B}(f)$ to denote the bandwidth of function $f$. We also use $\|\widehat{f}\|_1$ to denote the $\ell_1$-norm of $f$'s Fourier transform,

$$\|\widehat{f}\|_1 \;=\; \int |\widehat{f}(\boldsymbol{\xi})| \, \mathrm{d}\boldsymbol{\xi} \tag{2}$$

which we call the *Fourier $\ell_1$-norm* of $f$. Fourier $\ell_1$-norm can be interpreted as the absolute volume under $f$'s Fourier transform, and is an approximate measure of $\widehat{f}$'s sparsity. Fourier $\ell_1$-norm is both scale and shift invariant, i.e. if we define $g(\mathbf{x}) = f(\mathbf{W}\mathbf{x} + \mathbf{b})$ for a real-valued $f$ and $\mathbf{W} \in \mathbb{R}^{r \times k}$ and $\mathbf{b} \in \mathbb{R}^r$ for some $r \le k$, then $\|\widehat{g}\|_1 = \|\widehat{f}\|_1$. Some other useful properties of Fourier transform are:

- *Synthesis:* $f(\mathbf{x}) = \int \widehat{f}(\boldsymbol{\xi}) \exp\left(2\pi i \boldsymbol{\xi}^T \mathbf{x}\right) \mathrm{d}\boldsymbol{\xi}$, which also implies $\|\widehat{f}\|_1 = f(0)$ if $\widehat{f}$ is real and non-negative.
- *Shift:* $\widehat{f_{\mathbf{b}}}(\boldsymbol{\xi}) = \exp(2\pi i \mathbf{b}^T \boldsymbol{\xi}) \widehat{f}(\boldsymbol{\xi})$ where $f_{\mathbf{b}}(\mathbf{x}) := f(\mathbf{x} - \mathbf{b})$, which implies $\|\widehat{f_{\mathbf{b}}}\|_1 = \|\widehat{f}\|_1$ and $\mathcal{B}(f_{\mathbf{b}}) = \mathcal{B}(f)$.
- *Derivative:* $\widehat{\nabla f}(\boldsymbol{\xi}) = 2\pi i \, \widehat{f}(\boldsymbol{\xi}) \, \boldsymbol{\xi}$, where $\nabla f$ denotes the gradient of $f$.
- *Isometry:* $\int f(\mathbf{x})\overline{g(\mathbf{x})} \, \mathrm{d}\mathbf{x} = \int \widehat{f}(\boldsymbol{\xi})\overline{\widehat{g}(\boldsymbol{\xi})} \, \mathrm{d}\boldsymbol{\xi}$ where $\overline{z}$ denotes the complex conjugate of $z$.
- *Convolution:* $\widehat{fg} = \widehat{f} \star \widehat{g}$ where $\star$ denotes the convolution operator i.e. $\widehat{f} \star \widehat{g}(\boldsymbol{\xi}) := \int \widehat{f}(\boldsymbol{\eta})\widehat{g}(\boldsymbol{\xi} - \boldsymbol{\eta}) \, \mathrm{d}\boldsymbol{\eta}$. Therefore, $\mathcal{B}(fg) \le \mathcal{B}(f) + \mathcal{B}(g)$ and $\|\widehat{fg}\|_1 \le \|\widehat{f}\|_1 \|\widehat{g}\|_1$.

## 4 A FOURIER-BASED GENERALIZATION BOUND

Consider a supervised learning task with $n$ training samples $(\mathbf{x}_i, y_i)_{i=1}^n$ and function space $\mathcal{F}$. We are interested in uniform convergence bounds on the generalization risk. A standard approach to bound the generalization risk is based on the notion of Rademacher complexity. Given samples $(\mathbf{x}_i, y_i)_{i=1}^n$, the empirical Rademacher complexity of $\mathcal{F}$ is defined as

$$\mathcal{R}_n^{\mathrm{emp}}(\mathcal{F}) := \mathbb{E}_{\boldsymbol{\sigma}}\left[ \sup_{f \in \mathcal{F}} \frac{1}{n} \sum_{i=1}^n \sigma_i f(\mathbf{x}_i) \right] \tag{3}$$

where $\sigma_i$'s are i.i.d. random variables uniformly distributed over $\{-1, +1\}$. In fact, the Rademacher complexity of $\mathcal{F}$ measures how well $\mathcal{F}$ can fit some random labels over input $\mathbf{x}_i$'s. The following result shows how to bound the generalization risk over $\mathcal{F}$ through its Rademacher complexity.

**Theorem 1** (Bartlett & Mendelson (2002)). *Consider a $\rho$-Lipschitz loss function $\ell(f(\mathbf{x}), y)$ bounded as $|\ell(z, y)| \le c$. Then, for any $\delta > 0$, with probability at least $1 - \delta$*

$$\forall f \in \mathcal{F}: \quad \mathbb{E}\left[ \ell(f(\mathbf{X}), Y) \right] - \frac{1}{n} \sum_{i=1}^n \ell(f(\mathbf{x}_i), y_i) \;\le\; 2\rho \mathcal{R}_n^{\mathrm{emp}}(\mathcal{F}) + 4c\sqrt{\frac{2\log(4/\delta)}{n}}. \tag{4}$$

Since the Rademacher complexity of norm-bounded linear functions can be appropriately bounded (Kakade et al., 2009), one can effectively apply Theorem 1 to bound generalization risk over norm-bounded linear functions. To use Theorem 1 in the Fourier domain, here we provide a Rademacher complexity bound for bandlimited functions with bounded Fourier $\ell_1$-norm. We apply the following Rademacher complexity bound to bound generalization risk for 2-layer neural networks in Section 5, and also to analyze the performance of gradient-based methods with sinusoidal activation functions in Section 6.

**Theorem 2.** *Consider function space $\mathcal{F} = \big\{ f : \mathbb{R}^k \to \mathbb{R} \;\; \text{s.t.} \;\; \mathcal{B}(f) \leq B, \|\widehat{f}\|_1 \leq V \big\}$ of $B$-bandlimited functions with $V$-bounded Fourier $\ell_1$-norm. Then, the empirical Rademacher complexity for samples $(\mathbf{x}_i, y_i)_{i=1}^n$ is bounded as*

$$\mathcal{R}_n^{\text{emp}}(\mathcal{F}) \leq V \sqrt{\frac{4k \log\big(64\, nB \max_i \|\mathbf{x}_i\|_2\big)}{n}}. \tag{5}$$

*Proof.* We defer the proof to the Appendix. □

**Corollary 1.** *Assume that $\|\mathbf{X}\|_2 \leq C$ holds almost surely and the loss function $\ell$ is $\rho$-Lipschitz. Then, for any $\delta > 0$ with probability at least $1 - \delta$ the following generalization bound holds for any $B$-bandlimited function $f$ with $V$-bounded Fourier $\ell_1$-norm:*

$$\mathbb{E}\big[\, \ell(f(\mathbf{X}), Y)\,\big] - \frac{1}{n}\sum_{i=1}^n \ell(f(\mathbf{x}_i), y_i) \;\leq\; O\bigg(\rho V \sqrt{\frac{k \log(nBC/\delta)}{n}}\bigg). \tag{6}$$

*Proof.* The corrollary is a direct result of applying the bound in Theorem 2 to Theorem 1. □

The above corollary bounds the generalization risk uniformly over all bandlimited $f$'s such that $\mathcal{B}(f) \leq B$ and $\|\widehat{f}\|_1 \leq V$. Next, we apply the above results to 2-layer neural networks.

## 5 APPLICATION OF THEOREM 2 TO 2-LAYER NEURAL NETWORKS

Consider a 2-layer neural network including $d$ neurons with activation function $\phi$ in the hidden layer (See Figure 1a). The output of this neural network is

$$f_{\mathbf{a},\mathbf{W},\mathbf{b}}(\mathbf{x}) = \mathbf{a}^T \phi(\mathbf{W}\mathbf{x} + \mathbf{b}). \tag{7}$$

If $\phi$ has bounded bandwidth and Fourier $\ell_1$-norm, we can apply Theorem 2 to bound the Rademacher complexity and hence generalization risk over the 2-layer neural network. Here, we use $\|\mathbf{W}\|_{2,\infty}$ to denote the maximum $\ell_2$-norm $\|\mathbf{w}_i\|_2$ among all rows of $\mathbf{W}$.

**Corollary 2.** *Let $\mathcal{F}_\phi = \big\{ f(\mathbf{x}) = \mathbf{a}^T \phi(\mathbf{W}\mathbf{x} + \mathbf{b}) : \|\mathbf{W}\|_{2,\infty} \leq W, \|\mathbf{a}\|_1 \leq A \big\}$ be the class of 2-layer neural networks where $\mathcal{B}(\phi) = B$ and $\|\widehat{\phi}\|_1 = V$. Then, the empirical Rademacher complexity of $\mathcal{F}_\phi$ for samples $(\mathbf{x}_i, y_i)_{i=1}^n$ is bounded as follows*

$$\mathcal{R}_n^{\text{emp}}(\mathcal{F}_\phi) \leq O\bigg(AV \sqrt{\frac{k \log\big(nBW \max \|\mathbf{x}_i\|_2\big)}{n}}\bigg). \tag{8}$$

*Proof.* We defer the proof to the Appendix. □

Notice that for bandlimited activation functions with bounded Fourier $\ell_1$-norm, the above generalization bound is increasing logarithmically with $\|\mathbf{W}\|_{2,\infty}$. For example, this result holds for sinusoidal activation $\phi(x) = \sin(2\pi x)$ where $\|\widehat{\phi}\|_1 = 1$, $\mathcal{B}(\phi) = 1$. On the other hand, the existing Rademacher complexity bounds which use only the Lipschitz constant of the activation function are linear in $\mathbf{W}$'s norm (Bartlett & Mendelson, 2002). Therefore, by exploiting the spectral properties of $\phi$, Corollary 2 results in a tighter generalization bound than the bounds using only the Lipschitz constant of $\phi$.

However, an unbounded function such as ReLU $\phi(x) = \max(x, 0)$ has an infinite Fourier $\ell_1$-norm. Therefore, Corollary 2 does not directly apply to these functions. The following theorem uses a boundedness assumption on input $\mathbf{X}$ to apply Theorem 2 to ReLU-type activation functions. Although the following bound is growing faster than logarithmically with $\mathbf{W}$'s norm, it introduces new capacity norms for 2-layer ReLU-based networks.

**Theorem 3.** *Suppose that $\phi_\alpha(x) = \max\{x, \alpha x\}$ where $\alpha \in [0, 1]$ is an arbitrary constant. Consider the pair of dual norms $(\| \cdot \|_p, \| \cdot \|_q)$ where $1 \leq p, q \leq \infty$ and $1/p + 1/q = 1$. Assume that $\|\mathbf{x}_i\|_p \leq C$ holds for all $\mathbf{x}_i$'s. Then, for $\mathcal{F}_{\phi_\alpha} = \big\{ f_{\mathbf{a},\mathbf{W}}(\mathbf{x}) = \mathbf{a}^T \phi_\alpha(\mathbf{W}\mathbf{x}) : \sum_{i=1}^d |a_i|\|\mathbf{w}_i\|_q \leq V \big\}$*

$$\mathcal{R}_n^{\text{emp}}(\mathcal{F}_{\phi_\alpha}) \leq O\bigg(VC \sqrt{\frac{k \log(nkC)}{n}}\bigg). \tag{9}$$

*Proof.* We relegate the proof to the Appendix. □

The above bound uses the complexity score $\sum_{i=1}^{d} |a_i| \|\mathbf{w}_i\|_q$ for each $f_{\mathbf{a},\mathbf{W}}(\mathbf{x}) = \mathbf{a}^T \phi_\alpha(\mathbf{W}\mathbf{x})$. We can rewrite this complexity score in the following way, which is an $\ell_{1,q}$-group norm on the product of weights for each path from the input nodes to the output node of the 2-layer neural network,

$$\chi_q(f_{\mathbf{a},\mathbf{w}}) = \sum_{i=1}^{d} \left( \sum_{j=1}^{k} (|a_i| |w_{i,j}|)^q \right)^{1/q}. \tag{10}$$

Here $w_{i,j}$ denotes the weight on the link from the $j$th node of the input layer to the $i$th node of the hidden layer. Based on the path-norm function defined at (Neyshabur et al., 2015a), we call $\chi_q(f_{\mathbf{a},\mathbf{W}})$ the group path norm. For $q = 1$, $\chi_1$-group path norm leads to the $\ell_1$-path norm for 2-layer neural networks. We can use group path norms as an additive regularization penalty to learn over 2-layer neural networks. In our numerical experiments, we test the performance of $\chi_2$-group path norm and $\ell_1$-path norm regularization to control the generalization risk over 2-layer neural networks.

## 6 FOURIER ANALYSIS OF GRADIENT-BASED METHODS FOR 2-LAYER NEURAL NETWORKS WITH SINE ACTIVATION

In this section, we apply Fourier analysis for a 2-layer neural network with sinusoidal activation. We aim to understand the connection between generalizibility of local minima found by gradient-based methods and spectral properties of the population distribution $P_{\mathbf{X},Y}$. As a simplifying assumption, let's assume that target variable $Y$ is a deterministic function $Y(\mathbf{x})$ of input $\mathbf{X}$, which we call the labeling scheme. In our analysis, we consider the squared-error loss $\ell(y, y') = (y - y')^2$.

We specifically ask this question: how can spectral properties of labeling scheme $Y(\mathbf{x})$ and population density function $P_{\mathbf{X}}(\mathbf{x})$ affect the generalization performance of a gradient-based method? To address this question, we use a similar strategy to the analysis performed in (Mei et al., 2016) by establishing generalization results for both the empirical risk and the gradient of empirical risk. First, we show that the bandwidth and Fourier $\ell_1$-norm for the local minima of the population risk can be bounded in terms of the bandwidth and Fourier $\ell_1$-norm of $Y(\mathbf{x})$ and $P_{\mathbf{X}}(\mathbf{x})$. Next, we establish a generalization result for the gradient of the empirical risk, proving that the gradient of empirical risk would stay close to the gradient of population risk given that $Y(\mathbf{x})$ has limited bandwidth and Fourier $\ell_1$-norm. These two results show that by assuming a labeling scheme with limited bandwidth and Fourier $\ell_1$-norm, the local minima found by the gradient descent (in general large-batch gradient descent) method will generalize well.

### 6.1 POPULATION RISK WITH SINUSOIDAL ACTIVATION

Consider $f_{\mathbf{a},\mathbf{W},\mathbf{b}}(\mathbf{x}) = \sum_{j=1}^{d} a_j \sin(2\pi \mathbf{w}_j^T \mathbf{x} + b_j)$ coming from a 2-layer neural network with $d$ sinusoidal hidden units. Given the labeling scheme $Y(\mathbf{x})$ the population risk will be

$$\mathbb{E}_{P_{\mathbf{X}}} \left[ \ell(f_{\mathbf{a},\mathbf{W},\mathbf{b}}(\mathbf{x}), Y(\mathbf{x})) \right] = \mathbb{E}_{P_{\mathbf{X}}} \left[ \left( Y(\mathbf{x}) - \sum_{j=1}^{d} a_j \sin(2\pi \mathbf{w}_j^T \mathbf{x} + b_j) \right)^2 \right], \tag{11}$$

where the expectation is according to the population density function $P_{\mathbf{X}}(\mathbf{x})$.

**Lemma 1.** *Consider the population risk in* (11). *Assume $\mathbf{w}_j$ satisfies $\forall i \neq j$: $\min\{\|\mathbf{w}_i - \mathbf{w}_j\|_2, \|\mathbf{w}_i + \mathbf{w}_j\|_2\} > \mathcal{B}(P_{\mathbf{X}})$. Then, if $(\mathbf{a}, \mathbf{W}, \mathbf{b})$ is assumed to be a local minimum of the population risk,*

$$|a_j| \leq 2 \left| \widehat{Y} \star \widehat{P_{\mathbf{X}}}(\mathbf{w}_j) \right|. \tag{12}$$

*Proof.* We defer the proof to the Appendix. □

Lemma 1 says that if the component $a_j \sin(2\pi \mathbf{w}_j^T \mathbf{x})$ becomes isolated for a local minimum, by which we mean there are no other component $a_i \sin(2\pi \mathbf{w}_i^T \mathbf{x})$ with $\min\{\|\mathbf{w}_i - \mathbf{w}_j\|_2, \|\mathbf{w}_i + \mathbf{w}_j\|_2\}$ less than $P_{\mathbf{X}}$'s bandwidth, then the value of $a_j$ for that local minimum is nicely bounded in terms of the population distribution. This result leads to the following Theorem which describes the Fourier properties of the local minima of the population risk.

**Theorem 4.** *Consider the minimization problem of the population risk* (11). *If a local minimum* $(\mathbf{a}^*, \mathbf{W}^*, \mathbf{b}^*)$ *satisfies the isolated components condition, i.e. for any two different* $i, j$ *we have* $\min\{\|\mathbf{w}_i^* - \mathbf{w}_j^*\|_2, \|\mathbf{w}_i^* + \mathbf{w}_j^*\|_2\} > 2\mathcal{B}(P_{\mathbf{X}})$, *then for the local minimum function* $f_{\mathbf{a}^*, \mathbf{W}^*, \mathbf{b}^*}$

- $\mathcal{B}(f_{\mathbf{a}^*, \mathbf{W}^*, \mathbf{b}^*}) \leq \mathcal{B}(Y) + \mathcal{B}(P_{\mathbf{X}})$,

- $\|\widehat{f}_{\mathbf{a}^*, \mathbf{W}^*, \mathbf{b}^*}\|_1 \leq 2 \|\widehat{Y}\|_1$.

*Proof.* We defer the proof to the Appendix. $\qquad\square$

Theorem 4 implies that the bandwidth of the local minima of the population risk is less than the sum of bandwidths for $Y$ and $P_{\mathbf{X}}$. Also, the Fourier $\ell_1$-norm for the local minima of the population distribution is bounded by twice the Fourier $\ell_1$-norm of $Y$.

**Remark 1.** *To apply Theorem 4, the bandwidth of* $P_{\mathbf{X}}$ *needs to be smaller than half the distance among* $\mathbf{w}_i^*$'s. *For example, suppose that* $\mathbf{X} \sim \mathcal{N}(\boldsymbol{\mu}, \sigma^2 \mathbf{I}_{k \times k})$ *has a multivariate Gaussian distribution with mean* $\boldsymbol{\mu}$ *and diagonal covariance matrix with standard deviation* $\sigma$. *Then, the above theorem shows that if for any* $i, j$ *we have* $\min\{\|\mathbf{w}_i^* - \mathbf{w}_j^*\|_2, \|\mathbf{w}_i^* + \mathbf{w}_j^*\|_2\} > 2C/\sigma$ *for some constant* $C$, *then*

- $\mathcal{B}(f_{\mathbf{a}^*, \mathbf{W}^*, \mathbf{b}^*}) \leq \mathcal{B}(Y) + O(\sqrt{k}/\sigma)$,

- $\|\widehat{f}_{\mathbf{a}^*, \mathbf{W}^*, \mathbf{b}^*}\|_1 \leq 2(1 + d \exp(-C^2/2)) \|\widehat{Y}\|_1$.

*Proof.* See the proof of Theorem 4 in the Appendix. $\qquad\square$

## 6.2 GENERALIZATION TO THE EMPIRICAL RISK

Theorem 4 characterizes the Fourier properties of the local minima for the population risk. However, we want to investigate the generalization performance of the local minima of the empirical risk defined for training samples $(\mathbf{x}_i, Y(\mathbf{x}_i))_{i=1}^n$ as

$$\frac{1}{n} \sum_{i=1}^n \ell\big(f_{\mathbf{a}, \mathbf{W}, \mathbf{b}}(\mathbf{x}_i), Y(\mathbf{x}_i)\big) = \frac{1}{n} \sum_{i=1}^n \left( Y(\mathbf{x}_i) - \sum_{j=1}^d a_j \sin(2\pi \mathbf{w}_j^T \mathbf{x}_i + b_j) \right)^2. \quad (13)$$

To address this question, note that the bandwidth and Fourier $\ell_1$-norm of the loss's gradient with respect to each $a_j$ are bounded in terms of the bandwidth and Fourier $\ell_1$-norm of $Y(\mathbf{x})$ as

$$\big\| \nabla_{a_j} \ell\big(\widehat{f_{\mathbf{a}, \mathbf{W}, \mathbf{b}}}(\mathbf{x}), Y(\mathbf{x})\big) \big\|_1 \leq \|\widehat{Y}\|_1 + \|\mathbf{a}\|_1, \quad (14)$$

$$\mathcal{B}\big( \nabla_{a_j} \ell\big(f_{\mathbf{a}, \mathbf{W}, \mathbf{b}}(\mathbf{x}), Y(\mathbf{x})\big) \big) \leq \mathcal{B}(Y) + 2\|\mathbf{W}\|_{2,\infty}. \quad (15)$$

We can apply Corollary 1 to show that not only the empirical risk uniformly converges to the population risk but also the gradient of the empirical risk will stay close to the gradient of the population risk.

**Corollary 3.** *Consider* $f_{\mathbf{a}, \mathbf{W}, \mathbf{b}}(\mathbf{x}) = \sum_{j=1}^d a_j \sin(\mathbf{w}_j^T \mathbf{x} + b_j)$ *and squared error loss* $\ell$. *Then, given that* $\|\mathbf{X}\|_2 \leq C$, *for any* $\delta > 0$ *with probability at least* $1 - \delta$ *we have*

$$\forall j, \mathbf{a}, \mathbf{W}, \mathbf{b} \text{ s.t. } \|\mathbf{a}\|_1 + \|\widehat{Y}\|_1 \leq V, \ 2\|\mathbf{W}\|_{2,\infty} + \mathcal{B}(Y) \leq B : \quad (16)$$

$$\left| \mathbb{E}\big[\nabla_{a_j} \ell\big(f_{\mathbf{a}, \mathbf{W}, \mathbf{b}}(\mathbf{X}), Y(\mathbf{X})\big)\big] - \frac{1}{n} \sum_{i=1}^n \big[\nabla_{a_j} \ell\big(f_{\mathbf{a}, \mathbf{W}, \mathbf{b}}(\mathbf{x}_i), Y(\mathbf{x}_i)\big)\big] \right| \leq O\Big(V \sqrt{\frac{k \log(nBC/\delta)}{n}}\Big).$$

*Proof.* The corollary is a direct result of Corollary (1) given (14) and (15). Note that the generalization bound holds with probability $1 - \delta$ for the derivative with respect to all $a_j$'s, since the bounds in (14) and (15) hold for all $j$'s. $\qquad\square$

We emphasize that to prove Theorem 4 we need to analyze the risk function's derivative only with respect to $a_j$'s. Hence, generalization of the empirical risk's gradient with respect to $a_j$'s, which is

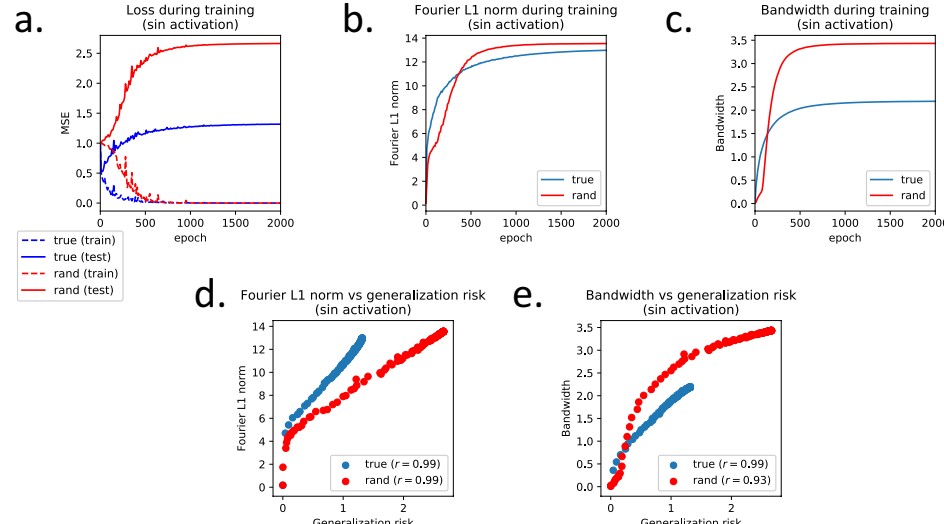

Figure 2: Training an test performance on cat and airplane CIFAR10 images with true and random labels. Sine activation and mean-squared-error loss were used.

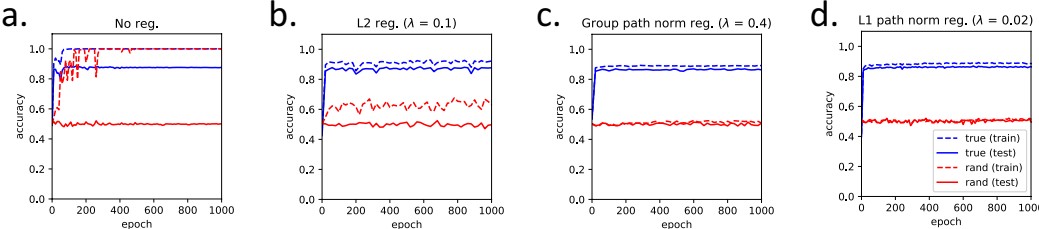

Figure 3: Training and test performance on cat and airplane CIFAR10 images with true and random labels. ReLU activation and cross-entropy loss were used.

shown in the above corollary under certain assumptions, is sufficient to apply an approximate version of Theorem 4 in section 8.6 to a local minimum $(\mathbf{a}^*, \mathbf{W}^*, \mathbf{b}^*)$ satisfying the isolated components assumption and found by the gradient descent approach initialized at a low $\|\mathbf{a}\|_1$ and $\|\mathbf{W}\|_{2,\infty}$. We can conclude that with probability at least $1 - \delta$ the $\ell_1$-norm of $f_{\mathbf{a}^*, \mathbf{W}^*, \mathbf{b}^*}$'s Fourier transform outside the bandwidth $\mathcal{B}(Y) + \mathcal{B}(P_{\mathbf{X}})$ is bounded by $O\big(dV\sqrt{\frac{k\log(nBC/\delta)}{n}}\big)$, and also

$$\|\widehat{f}_{\mathbf{a}^*, \mathbf{W}^*, \mathbf{b}^*}\|_1 \leq 2\|\widehat{Y}\|_1 + O\big(dV\sqrt{\frac{k\log(nBC/\delta)}{n}}\big).$$

Based on the above discussion, if a large-batch gradient descent method starts learning from $f_{\mathbf{a}, \mathbf{W}, \mathbf{b}}$ with low $\|\mathbf{a}\|_1$ and $\|\mathbf{W}\|_{2,\infty}$ and also we assume that the bandwidth and the Fourier $\ell_1$-norm for $Y(\mathbf{x})$ are properly bounded, Theorem 4 combined with Corollary 1 will guarantee good generalization performance for the local minima found by the gradient descent method.

## 7 NUMERICAL EXPERIMENTS

For all experiments described in this section, we implemented and trained the two-layer neural network described in Figure 1a using TensorFlow 1.3.0. We used SGD to train the model for 2000 epochs with an initial learning rate of 0.01. The learning rate decayed slightly each epoch at a rate of 0.95 every 390 epochs. We used $h = 512$ hidden units and a batch size of 128. When working with CIFAR10 data, we preprocessed the data as described in (Zhang et al., 2016), resulting in each training sample having dimension $d = 2352$. Initial weights from the first layer were sampled from $\mathcal{N}(0, 0.01/d)$ and initial weights from the second layer were sampled from $\mathcal{N}(0, 0.01/h)$.

## 7.1 SGD GRADUALLY LEARNS HIGHER FOURIER $\ell_1$-NORM, BANDWIDTH HYPOTHESES

We first numerically demonstrate that how Fourier $\ell_1$-norm and bandwidth both increases during training via SGD. Motivated by the analysis from Section 6, we use the squared-error as our loss function and sine as our activation function. Our samples consist of cats and airplanes from the CIFAR10 dataset with the labels mapped to $-1$ and $1$. We use 5000 and 2000 samples from each category for training and test, respectively. We arbitrarily chose two of the ten classes to accommodate our choice of loss function. We evaluate the network's performance for both random and true labels.

Figure 2a shows that without regularization, SGD learns to perfectly fit both the true and random labels, which is consistent with the results from Zhang et al. (2016). Additionally, the random labels are harder to learn, requiring more epochs before achieving a perfect fit. Figures 2b and 2c confirm that both Fourier $\ell_1$-norm and bandwidth consistently increase with training, highlighting how SGD gradually finds more complex hypotheses in order to fit the data. Finally, we see in figures 2d and 2e how both Fourier $\ell_1$-norm and bandwidth increase with generalization risk (the difference between test mean squared-error (MSE) and training MSE) with almost perfect correlation. This suggests that, as implied by the theory above, regularizing Fourier $\ell_1$-norm and bandwidth could improve generalizability of the final learned model.

## 7.2 GROUP PATH NORM REGULARIZATION FOR RELU ACTIVATION

We regularize group path norm for ReLU activation as motivated by Theorem 3. Although $\chi_2$-group path norm is not convex, it is differentiable and we can use it as an additive penalty and find a local minimum via SGD. Using the same experimental setup as from section 7.1, we swap sine for ReLU and test the network's performance for both random and true labels.

Figure 3a confirms that, like before, the network can fit both true and random labels. The generalization gap, however, remains large for random labels. By regularizing the $\ell_2$-norm of all the weights, we see that the generalization gap closes for both the true labels and the random labels without compromising test accuracy significantly (Figure 3b). This result is further improved when we use the $\chi_2$-group path norm and $\ell_1$-path norm (Figure 3c and 3d), demonstrating that direct regularization of Fourier $\ell_1$-norm leads to better generalization.

We cross-validated the value of $\lambda$ for each regularization technique, and we chose the $\lambda$ that resulted in the smallest generalization gap with comparable validation performance. To fairly compare different regularization strategies, we tested five lambda values for each strategy and then reported the performance on the test set for the lambda value that resulted in the best performance on the validation set.

We repeated the experiment using all 50000 CIFAR10 training samples (and 10000 test samples). We included all 10 classes and switched to cross-entropy loss. The results are shown in Figure 1b. Again, we see that while all regularization techniques give similar test performance, the generalization gap is closed significantly for the $\chi_2$-group path norm and $\ell_1$-path norm.

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

# 8 APPENDIX

## 8.1 PROOF OF THEOREM 2

We use a high-dimensional grid in the Fourier domain to approximate the Fourier transform of a $B$-bandlimited function. Consider the ball $\{\boldsymbol{\xi} : \|\boldsymbol{\xi}\|_2 \leq B\}$. Using the bounds on the covering number for $\ell_2$-norm, for any $0 < \epsilon < B$ we can find a set of points $\{\boldsymbol{\xi}_j : 1 \leq j \leq (3B/\epsilon)^k\}$ such that for any $\boldsymbol{\xi}$ with $\|\boldsymbol{\xi}\|_2 \leq B$, there exists some $\boldsymbol{\xi}_j$ with $\|\boldsymbol{\xi} - \boldsymbol{\xi}_j\|_2 \leq \epsilon$.

Let $S_j = \{\boldsymbol{\xi} : \|\boldsymbol{\xi} - \boldsymbol{\xi}_j\|_2 \leq \epsilon\}$ for each $1 \leq j \leq (3B/\epsilon)^k$. Note that $\{\boldsymbol{\xi} : \|\boldsymbol{\xi}\|_2 \leq B\} \subset \cup_j S_j$. We then define $S_j' = S_j \setminus \cup_{t=1}^{j-1} S_t$ to have a group of disjoint sets $S_j'$ covering $\{\boldsymbol{\xi} : \|\boldsymbol{\xi}\|_2 \leq B\}$. Since any $f \in \mathcal{F}$ is assumed to be $B$-bandlimited, for $f \in \mathcal{F}$

$$f(\mathbf{x}) = \int \widehat{f}(\boldsymbol{\xi}) \exp(2\pi i \boldsymbol{\xi}^T \mathbf{x}) \, \mathrm{d}\boldsymbol{\xi}$$

$$= \sum_{j=1}^{(3B/\epsilon)^k} \int_{\boldsymbol{\xi} \in S_j'} \widehat{f}(\boldsymbol{\xi}) \exp(2\pi i \boldsymbol{\xi}^T \mathbf{x}) \, \mathrm{d}\boldsymbol{\xi}. \tag{17}$$

Then, for any $f \in \mathcal{F} = \{f : \mathcal{B}(f) \leq B, \|\widehat{f}\|_1 \leq V\}$ we have

$$\left| f(\mathbf{x}) - \sum_{j=1}^{(3B/\epsilon)^k} \left[ \exp(2\pi i \boldsymbol{\xi}_j^T \mathbf{x}) \int_{\boldsymbol{\xi} \in S_j'} \widehat{f}(\boldsymbol{\xi}) \, \mathrm{d}\boldsymbol{\xi} \right] \right|$$

$$\stackrel{(a)}{=} \left| \sum_{j=1}^{(3B/\epsilon)^k} \int_{\boldsymbol{\xi} \in S_j'} \widehat{f}(\boldsymbol{\xi}) \left[ \exp(2\pi i \boldsymbol{\xi}^T \mathbf{x}) - \exp(2\pi i \boldsymbol{\xi}_j^T \mathbf{x}) \right] \mathrm{d}\boldsymbol{\xi} \right|$$

$$\leq \sum_{j=1}^{(3B/\epsilon)^k} \int_{\boldsymbol{\xi} \in S_j'} \left| \widehat{f}(\boldsymbol{\xi}) \left[ \exp(2\pi i \boldsymbol{\xi}^T \mathbf{x}) - \exp(2\pi i \boldsymbol{\xi}_j^T \mathbf{x}) \right] \right| \mathrm{d}\boldsymbol{\xi}$$

$$\stackrel{(b)}{\leq} \sum_{j=1}^{(3B/\epsilon)^k} \int_{\boldsymbol{\xi} \in S_j'} \left| \widehat{f}(\boldsymbol{\xi}) \right| 2\pi \|\mathbf{x}\|_2 \|\boldsymbol{\xi} - \boldsymbol{\xi}_j\|_2 \, \mathrm{d}\boldsymbol{\xi}$$

$$\leq 2\pi \|\mathbf{x}\|_2 \sum_{j=1}^{(3B/\epsilon)^k} \int_{\boldsymbol{\xi} \in S_j'} \left| \widehat{f}(\boldsymbol{\xi}) \right| \|\boldsymbol{\xi} - \boldsymbol{\xi}_j\|_2 \, \mathrm{d}\boldsymbol{\xi}$$

$$\stackrel{(c)}{\leq} 2\pi \epsilon \|\mathbf{x}\|_2 \sum_{j=1}^{(3B/\epsilon)^k} \int_{\boldsymbol{\xi} \in S_j'} \left| \widehat{f}(\boldsymbol{\xi}) \right| \mathrm{d}\boldsymbol{\xi}$$

$$= 2\pi \epsilon \|\mathbf{x}\|_2 \int \left| \widehat{f}(\boldsymbol{\xi}) \right| \mathrm{d}\boldsymbol{\xi}$$

$$\leq 2\pi \epsilon \|\mathbf{x}\|_2 V.$$

Here, (a) is a direct application of (17). (b) holds as $\exp(ibz) = \cos(bz) + i \sin(bz)$ is $b$-Lipschitz as a function of $z \in \mathbb{R}$ for any real number $b > 0$. (c) holds because according to our definitions $S_j' \subseteq S_j$ and $S_j = \{\boldsymbol{\xi} : \|\boldsymbol{\xi} - \boldsymbol{\xi}_j\|_2 \leq \epsilon\}$.

Therefore, the following function space $\mathcal{F}_\epsilon$ can approximate any $f \in \mathcal{F} = \{f : \mathcal{B}(f) \leq B, \|\widehat{f}\|_1 \leq V\}$ within $2\pi\epsilon CV$ accuracy for any $\|\mathbf{x}\|_2 \leq C$. Here $\mathbf{a}$ is, in general, a vector of complex numbers, and $\|\mathbf{a}\|_1 := \sum_j |a_j|$

where $|z|$ denotes the absolute value of complex number $z$,

$$\mathcal{F}_\epsilon = \left\{ f(\mathbf{x}) = \sum_{j=1}^{(3B/\epsilon)^k} a_j \exp(2\pi i \boldsymbol{\xi}_j^T \mathbf{x}) : \ \|\mathbf{a}\|_1 \leq V \right\}. \tag{18}$$

Then, $\mathcal{F}_\epsilon$ is the space of $\ell_1$-norm bounded linear functions in terms of the input vector $\left[\exp(2\pi i \boldsymbol{\xi}_j^T \mathbf{x})\right]_j$. Now, we can apply a well-known bound (Shalev-Shwartz & Ben-David, 2014) on the Rademacher complexity of $\ell_1$-norm bounded linear space $\mathcal{F}_{\text{lin},1} = \left\{ f : \mathbb{R}^k \to \mathbb{R} \ \text{s.t.} \ f(\mathbf{x}) = \mathbf{a}^T \mathbf{x}, \|\mathbf{a}\|_1 \leq A \right\}$ as

$$\mathcal{R}_n^{\text{emp}}(\mathcal{F}_{\text{lin},1}) \leq A \max_i \|\mathbf{x}_i\|_\infty \sqrt{\frac{2\log(2k)}{n}}. \tag{19}$$

Applying the above bound, we can bound the Rademacher complexity of $\mathcal{F}_\epsilon$ as

$$\mathcal{R}_n^{\text{emp}}(\mathcal{F}_\epsilon) \leq V \sqrt{\frac{2k \log(6B/\epsilon)}{n}}. \tag{20}$$

Since for each $f \in \mathcal{F}$ there exists $\tilde{f} \in \mathcal{F}_\epsilon$ such that $\forall \|\mathbf{x}\|_2 \leq C : \ |f(\mathbf{x}) - \tilde{f}(\mathbf{x})| \leq 2\pi\epsilon CV$,

$$\begin{aligned}
\mathcal{R}_n^{\text{emp}}(\mathcal{F}) &= \mathbb{E}_{\boldsymbol{\sigma}} \left[ \sup_{f \in \mathcal{F}} \frac{1}{n} \sum_{i=1}^n \sigma_i f(\mathbf{x}_i) \right] \\
&\leq \mathbb{E}_{\boldsymbol{\sigma}} \left[ \sup_{\tilde{f} \in \mathcal{F}_\epsilon} \frac{1}{n} \sum_{i=1}^n \sigma_i \tilde{f}(\mathbf{x}_i) \right] + 2\pi\epsilon V \max_i \|\mathbf{x}_i\|_2 \\
&= \mathcal{R}_n^{\text{emp}}(\mathcal{F}_\epsilon) + 2\pi\epsilon V \max_i \|\mathbf{x}_i\|_2.
\end{aligned} \tag{21}$$

Finally, combining (20) and (21) we obtain:

$$\forall \epsilon > 0 : \quad \mathcal{R}_n^{\text{emp}}(\mathcal{F}) \leq V \sqrt{\frac{2k \log(6B/\epsilon)}{n}} + 2\pi\epsilon V \max_i \|\mathbf{x}_i\|_2. \tag{22}$$

If we choose the value $\epsilon = \frac{1}{2\pi n \max_i \|\mathbf{x}_i\|_2}$, then we get

$$\begin{aligned}
\mathcal{R}_n^{\text{emp}}(\mathcal{F}) &\leq V \left( \sqrt{\frac{2k \log\left(12\pi nB \max_i \|\mathbf{x}_i\|_2\right)}{n}} + \frac{1}{n} \right) \\
&\leq V \sqrt{\frac{4k \log\left(12\pi nB \max_i \|\mathbf{x}_i\|_2\right) + 2/n}{n}} \\
&\leq V \sqrt{\frac{4k \log\left(64\, nB \max_i \|\mathbf{x}_i\|_2\right)}{n}},
\end{aligned} \tag{23}$$

where the last inequality follows from the fact that $1 \leq k, n$. Therefore, the proof is complete.

## 8.2 PROOF OF COROLLARY 2

First, we prove the following lemma.

**Lemma 2.** *Given function $f : \mathbb{R}^k \to \mathbb{R}$ and matrix $\mathbf{W} \in \mathbb{R}^{k \times k}$, we define $g(\mathbf{x}) = f(\mathbf{W}\mathbf{x})$. Then,*

- $\mathcal{B}(g) \leq \|\mathbf{W}\|_2 \mathcal{B}(f)$ *with $\|\mathbf{W}\|_2$ denoting the spectral norm of $\mathbf{W}$,*

- $\|\widehat{g}\|_1 = \|\widehat{f}\|_1$.

*Proof.* From the properties of the Fourier transform we know

$$\widehat{g}(\boldsymbol{\xi}) = \frac{1}{|\det(\mathbf{W})|} \widehat{f}(\mathbf{W}^{-T} \boldsymbol{\xi}). \tag{24}$$

Therefore, $\widehat{g}(\mathbf{W}^T\boldsymbol{\xi}') = \frac{1}{|\det(\mathbf{W})|}\widehat{f}(\boldsymbol{\xi}')$ and if $\|\boldsymbol{\xi}'\|_2 \leq \mathcal{B}(f)$, then $\|\mathbf{W}^T\boldsymbol{\xi}'\|_2 \leq \|\mathbf{W}\|_2\mathcal{B}(f)$ gives an upper-bound on $\mathcal{B}(g)$. Also,

$$
\begin{aligned}
\|\widehat{g}\|_1 &= \int |\widehat{g}(\boldsymbol{\xi})|\,\mathrm{d}\boldsymbol{\xi} \\
&= \int \frac{1}{|\det(\mathbf{W})|}|\widehat{f}(\mathbf{W}^{-T}\boldsymbol{\xi}))|\,\mathrm{d}\boldsymbol{\xi} \\
&= \frac{1}{|\det(\mathbf{W})|}\int |\widehat{f}(\mathbf{W}^{-T}\boldsymbol{\xi})|\,\mathrm{d}\boldsymbol{\xi} \\
&= \frac{1}{|\det(\mathbf{W})|}\int |\widehat{f}(\boldsymbol{\xi}'))|\frac{1}{|\det(\mathbf{W}^{-T})|}\,\mathrm{d}\boldsymbol{\xi}' \\
&= \int |\widehat{f}(\boldsymbol{\xi}')|\,\mathrm{d}\boldsymbol{\xi}' \\
&= \|\widehat{f}\|_1.
\end{aligned}
\tag{25}
$$

$\square$

It can be seen that this result remains valid even if $\mathbf{W}$ is not an invertible matrix, which will complete the proof for Corollary 2. However, we continue proving Corollary 2 without using this fact.

As shown in the above lemma, Fourier $\ell_1$-norm and bandwidth are invariant to an orthonormal transformation $\mathbf{W}$. Given $f_i(\mathbf{x}) = a_i\phi(\mathbf{w}_i^T\mathbf{x})$, we define $g_i(\mathbf{x}) = f_i(\mathbf{A}_i\mathbf{x})$ where $A_i$ is an orthonormal matrix with $\mathbf{w}_i$ as an eigenvector. Note that $\|\widehat{f_i}\|_1 = \|\widehat{g_i}\|_1$ and $\mathcal{B}(f_i) = \mathcal{B}(g_i)$. However, $g_i(\mathbf{x})$ is a function of only one of the coordinates, which we can assume, without loss of generality, to be the first coordinate. Hence, $g_i(\mathbf{x}) = a_i\phi(\|\mathbf{w}_i\|_2 x_1)$ for the first coordinate $x_1$, implying $\widehat{g_i}(\boldsymbol{\xi}) = \frac{a_i}{\|\mathbf{w}_i\|_2}\widehat{\phi}(\frac{\xi_1}{\|\mathbf{w}_i\|_2}).\boldsymbol{\delta}_2(\xi_2)\ldots\boldsymbol{\delta}_k(\xi_k)$ where $\boldsymbol{\delta}_j$ is the Dirac delta function across the $j$th dimension. Hence, we can use the above lemma in the 1-dimensional case to show $\|\widehat{g_i}\|_1 = |a_i|\|\widehat{\phi}\|_1$ and $\mathcal{B}(g_i) = \|\mathbf{w}_i\|_2\mathcal{B}(\phi)$. As a result,

$$\|\widehat{f_i}\|_1 = |a_i|\|\widehat{\phi}\|_1, \quad \mathcal{B}(f_i) \leq \|\mathbf{w}_i\|_2\mathcal{B}(\phi). \tag{26}$$

Hence, for $f(\mathbf{x}) = \mathbf{a}^T\phi(\mathbf{W}\mathbf{x} + \mathbf{b}) = \sum_{i=1}^d a_i\phi(\mathbf{w}_i^T\mathbf{x} + b_i)$ we have

$$\|\widehat{f}\|_1 \leq \|\mathbf{a}\|_1\|\widehat{\phi}\|_1, \quad \mathcal{B}(f) \leq \|\mathbf{W}\|_{2,\infty}\mathcal{B}(\phi). \tag{27}$$

The corollary is then a direct application of Theorem 2.

## 8.3 PROOF OF THEOREM 3

Given a ReLU-type activation function $\phi_\alpha(z) = \max\{z, \alpha z\}$,

$$\phi_\alpha(\mathbf{w}^T\mathbf{x}) = \|\mathbf{w}\|_q\phi_\alpha\big((\frac{\mathbf{w}}{\|\mathbf{w}\|_q})^T\mathbf{x}\big). \tag{28}$$

Since $\big\|\frac{\mathbf{w}}{\|\mathbf{w}\|_q}\big\|_q = 1$, if $\|\mathbf{x}\|_p \leq C$, then $\big|(\frac{\mathbf{w}}{\|\mathbf{w}\|_q})^T\mathbf{x}\big| \leq C$ and hence the input to $\phi_\alpha$ in the R.H.S. of (28) is always between $-C$ and $C$.

Suppose that function $\psi_\alpha$ satisfies $\psi_\alpha(z) = \phi_\alpha(z)$ for $z \in [-C, C]$. Then, based on the above discussion, we can bound the Rademacher complexity of $\mathcal{F}_{\phi_\alpha}$ by finding a bound on the Rademacher complexity of $\mathcal{F}_{\psi_\alpha} = \big\{f_{\mathbf{v},\mathbf{U}}(\mathbf{x}) = \mathbf{v}^T\psi_\alpha(\mathbf{U}\mathbf{x}) : \|\mathbf{v}\|_1 \leq V, \forall i : \|\mathbf{u}_i\|_q = 1\big\}$.

To find a good candidate for $\psi_\alpha$, we use a symmetrization trick to define

$$
\psi_\alpha(z) = \begin{cases}
-\alpha C & \text{if } z < -C, \\
\phi_\alpha(z) & \text{if } -C \leq z < C, \\
\phi_\alpha(2C - z) & \text{if } C \leq z < 3C, \\
-\alpha C & \text{if } 3C \leq z.
\end{cases}
\tag{29}
$$

Note that $\psi_\alpha(z) = (1 - \alpha)C\,h(\frac{z-C}{C}) + 2\alpha C\,h(\frac{z-C}{2C}) - \alpha C$ where $h(z) = \max\{0, 1 - |z|\}$. It can be seen that $\widehat{h}(\xi) = \big(\frac{\sin(\pi\xi)}{\pi\xi}\big)^2$ which is real and positive everywhere. Therefore, $\|\widehat{h}\|_1 = h(0) = 1$ which means that $\|\widehat{\psi_\alpha}\|_1 \leq C(1 + 2\alpha) \leq 3C$.

Since $|\widehat{h}(\xi)| \leq \frac{1}{\xi^2}$, we have $|\widehat{\psi_\alpha}(\xi)| \leq \frac{1}{\xi^2}$. For $B > 0$, we let the $B$-filtered $\psi_{\alpha,B}$ be a function with the following Fourier transform:

$$
\widehat{\psi_{\alpha,B}}(\xi) = \begin{cases}
\widehat{\psi_\alpha}(\xi) & \text{if } |\xi| \leq B \\
0 & \text{otherwise.}
\end{cases}
\tag{30}
$$

Then, since $\left|\widehat{\psi_\alpha}(\xi)\right| \leq \frac{1}{\xi^2}$ we have

$$\forall z \in \mathbb{R}: \quad \left|\psi_\alpha(z) - \psi_{\alpha,B}(z)\right| \leq \int_{|\xi| \geq B} \left|\widehat{\psi_\alpha}(\xi)\right| d\xi \leq \frac{2}{B}. \tag{31}$$

Thus, for any $B > 0$ the defined $\psi_{\alpha,B}$ approximates $\phi_\alpha$ with a maximum error of $\frac{2}{B}$ uniformly over $[-C, C]$. $\psi_{\alpha,B}$ also satisfies $\|\widehat{\psi_{\alpha,B}}\|_1 \leq 3C$ and $\mathcal{B}(\psi_{\alpha,B}) = B$. Applying Corollary 2, we get

$$\forall B > 0: \quad \mathcal{R}_n^{\text{emp}}(\mathcal{F}_{\phi_\alpha}) \leq O\left(VC\sqrt{\frac{k \log\left(nB \max \|\mathbf{x}_i\|_2\right)}{n}}\right) + \frac{2}{B}. \tag{32}$$

Here we can bound $\max_i \|\mathbf{x}_i\|_2 \leq \sqrt{k} \max_i \|\mathbf{x}_i\|_\infty \leq \sqrt{k}C$, and choose $B = n$ to get

$$\mathcal{R}_n^{\text{emp}}(\mathcal{F}_{\phi_\alpha}) \leq O\left(VC\sqrt{\frac{k \log\left(nkC\right)}{n}} + \frac{1}{n}\right), \tag{33}$$

which completes the proof.

## 8.4   Proof of Lemma 1

Note that
$$\nabla_{a_j} \mathbb{E}_{P_{\mathbf{X}}}\left[\ell\left(f_{\mathbf{a},\mathbf{W},\mathbf{b}}(\mathbf{X}), Y(\mathbf{X})\right)\right] = \mathbb{E}_{P_{\mathbf{X}}}\left[\nabla_{a_j} \ell\left(f_{\mathbf{a},\mathbf{W},\mathbf{b}}(\mathbf{X}), Y(\mathbf{X})\right)\right]$$
$$= \mathbb{E}_{P_{\mathbf{X}}}\left[\nabla_{a_j}\left(f_{\mathbf{a},\mathbf{W},\mathbf{b}}(\mathbf{X}) - Y(\mathbf{X})\right)^2\right]$$
$$= \mathbb{E}_{P_{\mathbf{X}}}\left[2 \sin(2\pi\mathbf{w}_j^T\mathbf{X} + b_j)\left(\sum_{t=1}^d a_t \sin(2\pi\mathbf{w}_t^T\mathbf{X} + b_t) - Y(\mathbf{X})\right)\right]$$
$$= \mathbb{E}_{P_{\mathbf{X}}}\left[2a_j \sin^2(2\pi\mathbf{w}_j^T\mathbf{X} + b_j)\right] - \mathbb{E}_{P_{\mathbf{X}}}\left[2 \sin(2\pi\mathbf{w}_j^T\mathbf{X} + b_j)Y(\mathbf{X})\right]$$
$$+ \mathbb{E}_{P_{\mathbf{X}}}\left[\sum_{t \neq j} 2a_t \sin(2\pi\mathbf{w}_t^T\mathbf{X} + b_t) \sin(2\pi\mathbf{w}_j^T\mathbf{X} + b_j)\right]$$
$$= a_j - 2\left[\cos(b_j) \operatorname{Im}\{\widehat{Y} \star \widehat{P_{\mathbf{X}}}(\mathbf{w}_j)\} + \sin(b_j) \operatorname{Re}\{\widehat{Y} \star \widehat{P_{\mathbf{X}}}(\mathbf{w}_j)\}\right]. \tag{34}$$

To show the last equality, we use the isolatedness assumption for $\mathbf{w}_j$, i.e. $\forall t \neq j: \min\{\|\mathbf{w}_t - \mathbf{w}_j\|_2, \|\mathbf{w}_t + \mathbf{w}_j\|_2\} > \mathcal{B}(P_{\mathbf{X}})$, and also $\|\mathbf{w}_j\|_2 \geq \mathcal{B}(P_{\mathbf{X}})/2$. Then, for each $t$
$$\mathbb{E}_{P_{\mathbf{X}}}\left[2 \sin(2\pi\mathbf{w}_t^T\mathbf{X} + b_t) \sin(2\pi\mathbf{w}_j^T\mathbf{X} + b_j)\right] = 2\int P_{\mathbf{X}}(\mathbf{x}) \sin(2\pi\mathbf{w}_t^T\mathbf{x} + b_t) \sin(2\pi\mathbf{w}_j^T\mathbf{x} + b_j) \, d\mathbf{x}$$
$$= \int P_{\mathbf{X}}(\mathbf{x})\left[\cos\left(2\pi(\mathbf{w}_t - \mathbf{w}_j)^T\mathbf{x} + b_t - b_j\right)\right.$$
$$\left. - \cos\left(2\pi(\mathbf{w}_t + \mathbf{w}_j)^T\mathbf{x} + b_t + b_j\right)\right] d\mathbf{x}$$
$$= 0.5 \exp(j(b_t - b_j))\widehat{P_{\mathbf{X}}}(\mathbf{w}_t - \mathbf{w}_j)$$
$$+ 0.5 \exp(j(b_j - b_t))\widehat{P_{\mathbf{X}}}(\mathbf{w}_j - \mathbf{w}_t)$$
$$- 0.5 \exp(j(b_t + b_j))\widehat{P_{\mathbf{X}}}(\mathbf{w}_t + \mathbf{w}_j)$$
$$- 0.5 \exp(-j(b_t + b_j))\widehat{P_{\mathbf{X}}}(-\mathbf{w}_t - \mathbf{w}_j)$$
$$= \begin{cases} 0 & \text{if } t \neq j, \\ 1 & \text{if } t = j. \end{cases} \tag{35}$$

Also, by applying the convolution property of Fourier transform we can show
$$\mathbb{E}_{P_{\mathbf{X}}}\left[\sin(2\pi\mathbf{w}_j^T\mathbf{X} + b_j)Y(\mathbf{X})\right] = \int P_{\mathbf{X}}(\mathbf{x})Y(\mathbf{x}) \sin(2\pi\mathbf{w}_j^T\mathbf{x} + b_j) \, d\mathbf{x}$$
$$= \int (P_{\mathbf{X}} \times Y)(\mathbf{x})\left[\cos(b_j) \sin(2\pi\mathbf{w}_j^T\mathbf{X}) + \sin(b_j) \cos(2\pi\mathbf{w}_j^T\mathbf{X})\right] d\mathbf{x}$$
$$= \cos(b_j) \operatorname{Im}\{\widehat{Y} \star \widehat{P_{\mathbf{X}}}(\mathbf{w}_j)\} + \sin(b_j) \operatorname{Re}\{\widehat{Y} \star \widehat{P_{\mathbf{X}}}(\mathbf{w}_j)\}.$$
Finally if $(\mathbf{a}, \mathbf{W}, \mathbf{b})$ is a local minimum for the population risk, for all $t$'s we have $\nabla_{a_t} \mathbb{E}_{P_{\mathbf{X}}}\left[\ell\left(f_{\mathbf{a},\mathbf{w},\mathbf{b}}(\mathbf{X}), Y(\mathbf{X})\right)\right] = 0$. Therefore, due to the isolatedness assumption of $\mathbf{w}_j$ we have

$$|a_j| = 2\left|\cos(b_j) \operatorname{Im}\{\widehat{Y} \star \widehat{P_{\mathbf{X}}}(\mathbf{w}_j)\} + \sin(b_j) \operatorname{Re}\{\widehat{Y} \star \widehat{P_{\mathbf{X}}}(\mathbf{w}_j)\}\right| \leq 2|\widehat{Y} \star \widehat{P_{\mathbf{X}}}(\mathbf{w}_j)|. \tag{36}$$

## 8.5 PROOF OF THEOREM 4

Since the isolatedness assumption holds for all $j$'s, by Lemma 1,

$$\forall j: \quad |a_j^*| \leq 2\big|\widehat{Y} \star \widehat{P_{\mathbf{X}}}(\mathbf{w}_j^*)\big|. \tag{37}$$

If $\|\mathbf{w}_t^*\|_2 > \mathcal{B}(Y) + \mathcal{B}(P_{\mathbf{X}})$ holds for some $t$, (37) implies

$$|a_t^*| \leq 2\big|\widehat{Y} \star \widehat{P_{\mathbf{X}}}(\mathbf{w}_t^*)\big| = 0. \tag{38}$$

Hence, $a_t^*$ will be 0, implying there will be no component in $f_{\mathbf{a}^*, \mathbf{W}^*, \mathbf{b}^*}$ with $\|\mathbf{w}_t^*\|_2 > \mathcal{B}(Y) + \mathcal{B}(P_{\mathbf{X}})$. This discussion proves the first part of Theorem, i.e. $\mathcal{B}(f_{\mathbf{a}^*, \mathbf{W}^*, \mathbf{b}^*}) \leq \mathcal{B}(Y) + \mathcal{B}(P_{\mathbf{X}})$.

To show the second part, note that

$$
\begin{aligned}
\|\widehat{f_{\mathbf{a}^*, \mathbf{W}^*, \mathbf{b}^*}}\|_1 &= \|\mathbf{a}^*\|_1 \\
&\overset{(a)}{\leq} 2 \sum_{t=1}^{d} \big|\widehat{Y} \star \widehat{P_{\mathbf{X}}}(\mathbf{w}_t^*)\big| \\
&= 2 \sum_{t=1}^{d} \left| \int \widehat{Y}(\boldsymbol{\xi}) \widehat{P_{\mathbf{X}}}(\mathbf{w}_t^* - \boldsymbol{\xi}) \, d\boldsymbol{\xi} \right| \\
&\leq 2 \sum_{t=1}^{d} \int \left| \widehat{Y}(\boldsymbol{\xi}) \widehat{P_{\mathbf{X}}}(\mathbf{w}_t^* - \boldsymbol{\xi}) \right| d\boldsymbol{\xi} \\
&= 2 \sum_{t=1}^{d} \int \big|\widehat{Y}(\boldsymbol{\xi})\big| \big|\widehat{P_{\mathbf{X}}}(\mathbf{w}_t^* - \boldsymbol{\xi})\big| \, d\boldsymbol{\xi} \\
&= 2 \int \big|\widehat{Y}(\boldsymbol{\xi})\big| \left[ \sum_{t=1}^{d} \big|\widehat{P_{\mathbf{X}}}(\mathbf{w}_t^* - \boldsymbol{\xi})\big| \right] d\boldsymbol{\xi} \\
&\overset{(b)}{\leq} 2 \sum_{t=1}^{d} \int \big|\widehat{Y}(\boldsymbol{\xi})\big| \, d\boldsymbol{\xi} \\
&= 2\|\widehat{Y}\|_1.
\end{aligned}
\tag{39}
$$

Here, (a) comes from Lemma 1. Also, since $\min\big\{\|\mathbf{w}_t^* - \mathbf{w}_r^*\|_2, \|\mathbf{w}_t^* + \mathbf{w}_r^*\|_2\big\} > 2\,\mathcal{B}(P_{\mathbf{X}})$ is assumed for any $t \neq r$, for any $\boldsymbol{\xi}$ at most one element in $\big[\widehat{P_{\mathbf{X}}}(\mathbf{w}_t^* - \boldsymbol{\xi})\big]_{t=1}^{d}$ can be nonzero. Because if both $\widehat{P_{\mathbf{X}}}(\mathbf{w}_t^* - \boldsymbol{\xi})$ and $\widehat{P_{\mathbf{X}}}(\mathbf{w}_r^* - \boldsymbol{\xi})$ are nonzero for $r \neq t$, then $\|\mathbf{w}_t^* - \boldsymbol{\xi}\| \leq \mathcal{B}(P_{\mathbf{X}})$ and also $\|\mathbf{w}_r^* - \boldsymbol{\xi}\| \leq \mathcal{B}(P_{\mathbf{X}})$ which results in $\|\mathbf{w}_t^* - \mathbf{w}_r^*\| \leq 2\mathcal{B}(P_{\mathbf{X}})$ which is a contradiction. Hence,

$$\sum_{t=1}^{d} \big|\widehat{P_{\mathbf{X}}}(\mathbf{w}_t^* - \boldsymbol{\xi})\big| \leq \max_{\boldsymbol{\xi}'} \big|\widehat{P_{\mathbf{X}}}(\boldsymbol{\xi}')\big| \leq \int \big|P_{\mathbf{X}}(\mathbf{x})\big| \, d\mathbf{x} = 1, \tag{40}$$

which proves (b) and completes the proof.

### 8.5.1 APPLYING THEOREM 4 TO MULTIVARIATE GAUSSIAN $\mathbf{X}$

Assume $\mathbf{X} \sim \mathcal{N}(\boldsymbol{\mu}, \sigma^2 \mathbf{I}_{k \times k})$ has a multivariate Gaussian distribution with mean $\boldsymbol{\mu}$ and standard deviation $\sigma$. Then, the Fourier transform $\hat{P}_{\mathbf{X}}$ has a Gaussian shape with mean $\mathbf{0}$ and standard deviation $1/\sigma$. Hence, if for any $i, j$ we have $\min\big\{\|\mathbf{w}_i^* - \mathbf{w}_j^*\|_2, \|\mathbf{w}_i^* + \mathbf{w}_j^*\|_2\big\} > 2C/\sigma$ for some constant $C$, the approximation error term which should be added to the upperbound in Equation (39) is $2\|\widehat{Y}\|_1 d \exp(-C^2/2)$. Also, given any $\epsilon > 0$ the Fourier $\ell_1$ norm outside the bandwidth $O(\sqrt{k}\log(1/\epsilon)/\sigma)$ is at most $\epsilon$. Therefore, Theorem 4 implies

- $\mathcal{B}(f_{\mathbf{a}^*, \mathbf{W}^*, \mathbf{b}^*}) \leq \mathcal{B}(Y) + O\big(\sqrt{k}/\sigma\big)$,
- $\|\widehat{f}_{\mathbf{a}^*, \mathbf{W}^*, \mathbf{b}^*}\|_1 \leq 2(1 + d \exp(-C^2/2)) \|\widehat{Y}\|_1$.

## 8.6 APPROXIMATE VERSION OF THEOREM 4

Here we show an approximate version of Theorem 4 which applies to approximate population local minima.

**Theorem 5.** *Consider minimizing the population risk* (11). *Consider an approximate local minimum* $(\mathbf{a}^*, \mathbf{W}^*, \mathbf{b}^*)$ *where* $\big|\nabla_{a_j}\mathbb{E}\big[\ell\big(f_{\mathbf{a}^*, \mathbf{W}^*, \mathbf{b}^*}(\mathbf{X}), Y(\mathbf{X})\big)\big)\big]\big| \leq \epsilon$ *for all $j$'s. If for any two different $i, j$ we*

*have* $\min\{\|\mathbf{w}_i^* - \mathbf{w}_j^*\|_2, \|\mathbf{w}_i^* + \mathbf{w}_j^*\|_2\} > 2\,\mathcal{B}(P_{\mathbf{X}})$, *then the Fourier* $\ell_1$-*norm of* $f_{\mathbf{a}^*,\mathbf{W}^*,\mathbf{b}^*}$ *outside the bandwidth* $\mathcal{B}(Y) + \mathcal{B}(P_{\mathbf{X}})$ *is bounded by* $d\,\epsilon$ *and*

$$\|\widehat{f}_{\mathbf{a}^*,\mathbf{W}^*,\mathbf{b}^*}\|_1 \leq 2\,\|\widehat{Y}\|_1 + d\,\epsilon.$$

*Proof.* Since the isolated components condition holds, we can apply Lemma 1's proof to show under the above assumptions

$$\forall\, j:\quad |a_j^*| \leq 2\big|\hat{Y} \star \hat{P}_{\mathbf{X}}(\mathbf{w}_j^*)\big| + \epsilon.$$

Then, a simple modification of Theorem 4's proof according to the above inequality proves the above theorem. □

## 8.7 PROOF OF THEOREM 4 WITHOUT THE ISOLATED COMPONENTS ASSUMPTION

What happens if a component $\mathbf{w}_i$ is not isolated from the other components which has been assumed in Theorem 4? As a simplifying assumption, we assume that $\mathbf{b} = \mathbf{0}$ and $\widehat{P_{\mathbf{X}}}$ is real. We can write

$$\forall\, i:\ \nabla_{a_i} \mathbb{E}_{P_{\mathbf{X}}}\big[\ell\big(f_{\mathbf{a},\mathbf{W}}(\mathbf{x}), Y(\mathbf{x})\big)\big] = \sum_{j=1}^{d}\big[a_j \widehat{P_{\mathbf{X}}}(\mathbf{w}_i - \mathbf{w}_j)\big] - \mathrm{Im}\big\{\hat{Y} \star \widehat{P_{\mathbf{X}}}(\mathbf{w}_i)\big\}$$

$$\approx \sum_{j=1}^{d}\bigg[\bigg(a_j - \int_{\boldsymbol{\xi} \in S_{\mathbf{w}_j}} \mathrm{Im}\{\widehat{Y}(\boldsymbol{\xi})\}\,\mathrm{d}\boldsymbol{\xi}\bigg)\widehat{P_{\mathbf{X}}}(\mathbf{w}_i - \mathbf{w}_j)\bigg] \tag{41}$$

Here $S_{\mathbf{w}_j}$'s, which are centered around $\mathbf{w}_j$'s, are disjoint sets covering the bandwidth region for $\widehat{Y}$, i.e. $\{\boldsymbol{\xi}:\ \|\boldsymbol{\xi}\|_2 \leq \mathcal{B}(Y)\} \subseteq \bigcup_j S_{\mathbf{w}_j}$. Note that in (41) we have approximated the convolution integral as

$$\widehat{Y} \star \widehat{P_{\mathbf{X}}}(\mathbf{w}_i) = \int_{\boldsymbol{\xi}:\ \|\boldsymbol{\xi}\|_2 \leq \mathcal{B}(Y)} \widehat{P_{\mathbf{X}}}(\mathbf{w}_i - \boldsymbol{\xi})\widehat{Y}(\boldsymbol{\xi})\,\mathrm{d}\boldsymbol{\xi}$$

$$= \sum_{j=1}^{d} \int_{\boldsymbol{\xi} \in S_{\mathbf{w}_j}} \widehat{P_{\mathbf{X}}}(\mathbf{w}_i - \boldsymbol{\xi})\widehat{Y}(\boldsymbol{\xi})\,\mathrm{d}\boldsymbol{\xi}$$

$$\approx \sum_{j=1}^{d} \widehat{P_{\mathbf{X}}}(\mathbf{w}_i - \mathbf{w}_j) \int_{\boldsymbol{\xi} \in S_{\mathbf{w}_j}} \widehat{Y}(\boldsymbol{\xi})\,\mathrm{d}\boldsymbol{\xi}.$$

Letting the gradient element in (41) be zero for all $a_i$'s at a local minimum $(\mathbf{a}^*, \mathbf{W}^*)$ of the population risk, the following approximation holds in general case:

$$\forall\, j:\ a_j^* \approx \int_{\boldsymbol{\xi} \in S_{\mathbf{w}_j}} \mathrm{Im}\{\widehat{Y}(\boldsymbol{\xi})\}\,\mathrm{d}\boldsymbol{\xi}, \tag{42}$$

Here, the matrix $\big[\widehat{P_{\mathbf{X}}}(\mathbf{w}_i - \mathbf{w}_j)\big]_{1 \leq i,j \leq d}$ is positive-definite and hence invertible, because $\widehat{P_{\mathbf{X}}}$ is the Fourier transform of $P_{\mathbf{X}}$ and due to Bochner's theorem a positive-definite kernel function. Therefore, the system of linear equations $\big[\widehat{P_{\mathbf{X}}}(\mathbf{w}_i^* - \mathbf{w}_j^*)\big]\big[a_j^* - \int_{\boldsymbol{\xi} \in S_{\mathbf{w}_j}} \mathrm{Im}\{\widehat{Y}(\boldsymbol{\xi})\}\,\mathrm{d}\boldsymbol{\xi}\big] \approx \mathbf{0}$ would imply (42). This discussion indicates that the result of Theorem 4 would remain valid even if the isolated components condition does not hold.

