# OpenReview forum: "A Spectral Approach to Generalization and Optimization in Neural Networks"
_ICLR.cc/2018/Conference — Reject_

### Official Review · AnonReviewer2 · 2017-11-15
**application domain seems restricted**

**Rating:** 6
**Confidence:** 3

**Review:**

Deep neural networks have found great success in various applications. This paper presents a theoretical analysis for 2-layer neural networks (NNs) through a spectral approach. Specifically, the authors develop a Fourier-based generalization bound. Based on this, the authors show that the bandwidth, Fourier l_1 norm and the gradient for local minima of the population risk can be controlled for 2-layer NNs with SINE activation functions. Numerical experimental results are also presented to verify the theory.

(1) The scope is a bit limited. The paper only considers 2-layer NNs. Is there an essential difficulty in extending the result here to NNs with more layers? Also, the analysis for gradient-based method in section 6  is only for squared-error loss, SINE activation and a deterministic target variable. What would happen if Y is random or the activation is ReLU?
(2) The generalization bound in Corollary 3 is only for the gradient w.r.t. \alpha_j. Perhaps, an object of more interest is the gradient w.r.t. W. It would be intersting to present some analysis regarding the gradient w.r.t. W.
(3) It is claimed that the bound is tighter than that obtained using only the Lipschitz property of the activation function. However, no comparison is clearly made. It would be better if the authors could explain this more?

In summary, the application domain of the theoretical results seems a bit restricted.

Minor comments:
Eq. (1): d\xi should be dx
Lemma 2: one \hat{g} should be \hat{f}

---

> ### Author Response · Authors · 2017-12-15
> **Author Response to AnonReviewer2**
>
> Thank you for your feedback. First let us note that analyzing the generalization performance of gradient methods is an open problem, even in the specific case of 2-layer neural nets and squared-error loss function. Our focus in this work is to address this open problem in the special case of 2-layer NNs with sine activation. We have chosen and analyzed this simple neural network model via Fourier analysis, because as shown in our numerical experiments (section 7) this simple model can still easily overfit random labels. Here is our response to the other questions and comments:
>
> 1)  A possible way of extending our Fourier-based analysis to multi hidden layer neural nets is through the analysis of the Fourier representation of composite of sine functions. To apply our Fourier-based analysis of gradient methods for ReLU function, one possible way is to approximate ReLU by its Fourier series after assuming the input X is properly bounded. For a random Y(x), we need to perform our Fourier analysis for the resulting stochastic process instead of a deterministic function.
>
> 2)  A sufficient condition for Theorem 4’s bounds is the generalization of the gradients w.r.t. \alpha. Generalization of the gradients only w.r.t. W does not provide a sufficient condition for applying Theorem 4. Also, establishing generalization w.r.t. both \alpha and W leads to slower rates of convergence compared to our result in Corollary 3.
>
> 3)  For a two layer neural net f(x)=a*phi(Wx) with activation phi, the Lipschitz-based bounds are linear in the square root of W’s norm. Our Corollary 2 improves this dependency to the square root of the log of W’s norm. The improvement holds for bandlimited phi’s such as sine or Gaussian activations. We will make this point clearer in the text.
>
> 4)  Thanks for pointing out the typos. We will correct them in our draft.

---

### Official Review · AnonReviewer3 · 2017-11-26

**Rating:** 6
**Confidence:** 3

**Review:**


This work proposes to study the generalization of learning neural networks via the Fourier-based method. It first gives a Fourier-based generalization bound, showing that Rademacher complexity of functions with small bandwidth and Fourier l_1 norm will be small. This leads to generalization for 2-layer networks with appropriate bounded size. For 2-layer networks with sine activation functions, assuming that the data distribution has nice spectral property (ie bounded bandwidth), it shows that the local minimum of the population risk (if with isolated component condition) will have small size, and also shows that the gradient of the empirical risk is close to that of the population risk. Empirical results show that the size of the networks learned on random labels are larger than those learned on true labels, and shows that a regularizer implied by their Fourier-based generalization bound can effectively reduce the generalization gap on random labels.

The idea of applying the Fourier-based method to generalization is interesting. However, the theoretical results are not very satisfactory.
-- How do the bounds here compared to those obtained by directly applying Rademacher complexity to the neural network functions?
-- How to interpret the isolated components condition in Theorem 4? Basically, it means that B(P_X) should be a small constant. What type of distributions of X will be a good example?
-- It is not easy to put together the conclusions in Section 6.1 and 6.2. Suppose SGD leads to a local minimum of the empirical loss. One can claim that this is an approximate local minimum (ie, small gradient) by Corollary 3. But to apply Theorem 4, one will need a version of Theorem 4 for approximate local minima. Also, one needs to argue that the local minimum obtained by SGD will satisfy the isolated component condition. The argument in Section 8.6 is not convincing, ie, there is potentially a large approximation error in (41) and one cannot claim that Lemma 1 and Theorem 4 are still valid without the isolated component condition.

---

> ### Author Response · Authors · 2017-12-15
> **Author Response to AnonReviewer3**
>
> Thank you for your feedback. Here is our response to the three points raised in this review:
>
> 1)  The existing Rademacher complexity bounds for neural nets are based on the activation function’s Lipschitz constant. For a two layer neural net f(x)=a*phi(Wx) with activation phi, those Lipschitz-based bounds are linear in the square root of W’s norm. Our Corollary 2 improves this dependency to the square root of the log of W’s norm. The improvement holds for bandlimited phi’s such as sine or Gaussian activations.
>
>   Although the ReLU function does not satisfy the above condition, our Fourier-based generalization result is still applicable to ReLU-based networks (Theorem 3) and motivates new capacity norms (group path norms) for these networks.
>
> 2)  For example, consider a multivariate Gaussian X ~ N( 0 , sigma^2*I). The Fourier transform of P_X in this case, which is  exp{ - (sigma*||w||)^2 / 2 }, becomes sufficiently small in Eq. (39) if the components are O( 1/sigma ) apart. Therefore, we need the standard deviation of X to be large enough so that the components are O( 1/sigma ) apart. We will add this example to section 6.
>
> 3)  For an approximate local minimum where the population gradient is epsilon instead of zero, the upper-bound in Lemma 1 (Eq. (12)) changes by at most epsilon and hence the Fourier L1_norm bound in Theorem 4 changes by at most d*epsilon (d is the size of the hidden layer).
>
>   We agree that the isolated components assumption provides a barrier for applying our theory to the general case. However, the condition holds given that Y(x)’s Fourier transform has distant local extrema (at least B(P_X) apart). For example, the condition holds if Y(x)=a^T sin(Wx) where each two different rows w_i,w_j of W satisfy || w_i - w_j || > B(P_X). We will include this discussion in section 6.

---

### Official Review · AnonReviewer1 · 2017-11-26

**Rating:** 4
**Confidence:** 4

**Review:**

This paper studies the generalization properties of 2-layer neural networks based on Fourier analysis. Studying the generalization property of neural network is an important problem and Fourier-based analysis is a promising direction, as shown in (Lee et al., 2017). However, I am not satisfied with the results in the current version.

1) The main theoretical results are on the sin activation functions instead of commonly used ReLU functions.

2) Even if for sin activation functions, the analysis is NOT complete. The authors claimed in the abstract that gradient-based methods will converge to generalizable local minima. However, Corollary 3 is only a concentration bound on the gradient. There is a gap that how this corollary implies generalization. The paragraph below this corollary is only a high level intuition.

---

> ### Author Response · Authors · 2017-12-15
> **Author Response to AnonReviewer1**
>
> Thanks for your feedback. Here is our response to the two comments:
>
> 1)  The Fourier-based generalization bound in Theorem 2 is applicable to a general activation function. By applying this generalization result, Theorem 3 motivates new capacity norms (group path norms) for ReLU-based networks. Supporting this, our numerical experiments in sections 7.2,7.3 indicate that regularizing group path norms can close the generalization gap without compromising test accuracy over neural nets with ReLU activation.
>
>   In section 6, we apply our Fourier-based generalization result to address the open problem of analyzing generalization performance of gradient methods in the special case of 2-layer neural nets with sine activation. Here we have chosen and analyzed this simple neural network model via Fourier analysis, because as shown in our numerical experiments this simple neural network can still easily overfit random labels.
>
> 2)  Corollary 3 is not the final generalization bound, but it should be applied together with Theorem 4. More specifically, Corollary 3 provides a sufficient condition to apply the bounds in Theorem 4 on the spectral properties of the local minima found, which in turn bounds the generalization error of those local minima.
>
>   The generalization result holds for the local minima found by large-batch gradient descent, and the paragraph after Corollary 3 gives high level intuition why we expect the bound to also hold for the local minima found by small-batch gradient descent. We will make this explanation clearer in the text.

---

### Public Comment · (anonymous) · 2017-11-14
**Theoretical results and experiments aren't consistent (sometimes contradicting).**

Summary: The authors propose 1) New class of activation functions which have bounded bandlimit (in the foruier domain) 2) and 'nice' l1-norm in the fourier domain. They evaluate the generalization bound derived by Bartlett and Mendelson (2002) for bandlimited functions and get  tighter gaurantees than just using lipschitz continuity. They also extend the analysis to the gradients of the loss functions. They also have a few experiments trying to support their claims.

A few concerns -

1) The authors propose using sin function as an activation function over ReLu (or sigmoid). However, they haven't directly compared the accuracy of NNs using ReLu with NNs using sin function. To be more specific, Figure 2 evaluate NNs with sin-activation function using MSE whereas Figure 3 evaluates NNs with ReLu-activation functions using prediction accuracy.

2) In the introduction, Keskar et. al. are cited claiming - 'SGD has been empirically shown to outperform large-batch gradient descent'. However, the Corollary 3 seems to say that the difference between sample-gradient and population-gradient is smaller for large values of samples and hence large-batch gradient descent should outperform SGD. The previous two statements are in direct contradiction.

3) The experimental section states that - "We note that while we tested multiple values of lambda for each regularization technique, we always chose lambda the that resulted in the smallest generalization gap with comparable test performance."
Such techniques are classic examples on how to overfit. The standard practice in the community is to choose the lambda via cross-validation.

4) In figure 1b the authors claim that path-norm penalty reduces the generalization error (compared l2 penalty). However, on close observation I see that using path-norm penalty also reduced the accuracy (which is not mentioned in the description). It is well known that reducing the size of the class of functions (to optimize over), reduces both training accuracy and generalization error. Hence it most likely that using path norm is 'effectively' reducing the class of functions and hence decreasing the generalization error (along with accuracy). One could quite possibly acheive the same effect by using l2 norm with a higher value of lambda.

5) From Fig 2b the authors conclude that random labels will have a higher path-norm. However, this is not very convincing to me as it looks like the path-norm for true labels might be equal to the path-norm for the random labels if the experiments were run for a few more epocs.


Few minor points
1) The introduction claims gaussian activation function as bandlimited. Can you explain this?
2) The assumption in lemma 1 are used only to prove the result and hence seems a little artificial. What is the interpretation of such an assumption?
3) Lemma 1 requires ||w_j|| to be larger the bandwith of P_X. Large values of W would make the sin function a highly non-monotonous activation function (with high frequency). I don't think such a highly variable non-monotonous functions is a good candidate for an activation functions.


Conclusion: There is huge disconnect between theretical conclusions and experiments results in this paper.
1) The theory proposes using sin-activation activation function - However, the sin-activation is not directly compared to ReLu in any experiments.
2) The theory proposes superiority of path-norm over l2 norm - The experiments are inconclusive (point 5).
3) The paper claims the gradient methods that accurately estimate the population gradient are expected to have better performance - Keskar et. al. (https://arxiv.org/pdf/1609.04836.pdf) have the exact opposite conclusion.

---

> ### Author Response · Authors · 2017-11-21
> **Numerical experiments support theoretical results**
>
> The following addresses the points made in the conclusion:
>
> 1) Sinusoidal activation is not proposed as a replacement for RELU, but as an analytical simplification to illustrate that gradient-based methods converge to generalizable local minima.
>
> 2) Our numerical results in Figures 1B and 3 indicate that group path norms can close the generalization gap for both random and true labels without compromising test accuracy for true labels, while L2_norm cannot.
>
> 3) Corollary 3 only applies to large-batch gradient descent and it does not explain the difference between the generalization performance of small and large batch gradient-descent.  Hence it does not contradict Keskar et al.’s results.
>
> Here is the response to the other concerns:
>
> 3) We have a typo in this sentence and the word “test” should be replaced with “validation.” To fairly compare different regularization strategies, we tested about 5 lambda values for each strategy and then reported the performance on the test set for the lambda value that resulted in the best performance on the validation set. Good performance here means low generalization gap with comparable validation accuracy for the true labels.
>
> 4) Due to computational constraints, we could only test a small set of lambda values for each regularization strategy. For each strategy, we chose the largest lambda coefficient which did not result in more than a 5% decrease in the validation accuracy for true labels. For the X2-group path norm (Figure 1B3), this resulted in test and train accuracies of 0.519 and 0.632 for the true labels and 0.099 and 0.104 for the random labels. For the L2-norm (Figure 1B2), this resulted in test and train accuracies of 0.540 and 0.659 for the true labels and 0.096 and 0.285 for the random labels. Therefore in comparison to the L2 norm, the X2-group path norm achieves smaller generalization gap for both true and random labels (significantly for random labels) without really compromising test accuracy.
>
> 5) The point of Figure 2b is to validate our Fourier-based generalization bound, and not to examine path norms which is irrelevant to the plots in Figure 2. Figure 2b demonstrates how the Fourier L1-norm, which is different from path norm, of a neural net with sine activation changes during training. The hypotheses fitting random and true labels have comparable Fourier L1 norm (slightly larger for random labels), but as shown in Figure 2c the bandwidth achieved when fitting random labels is > 1.5 times larger than the bandwidth achieved when fitting true labels. Our generalization result in Theorem 2 depends on both Fourier L1-norm and bandwidth, which correctly predicts that the generalization risk should be larger for random labels.
>
> Response to the minor points:
>
> 1) The Fourier transform of a Gaussian function has a Gaussian shape. Therefore, a Gaussian function’s Fourier transform is concentrated around the origin in a ball with radius inversely proportional to the standard deviation of that Gaussian function and can be arbitrarily well approximated by a bandlimited function.
>
> 2,3) This is a technical assumption to obtain the exact value instead of an approximation of a convolution integral. As we have discussed in the Appendix (section 8.6), Theorem 4 (shown through Lemma 1) remains approximately valid even without the isolated components assumption.

---

> > ### Public Comment · (anonymous) · 2017-11-23
> > **Points not addressed satisfactorily.**
> >
> > 1) Simplifying ReLU by sinusoidal function doesn't seem like a good idea and seems very forced.
> >
> > 2) Comment (2) directly contradicts the numbers in comment (4). Refer to my point 4.
> >
> > 3) Since large-batch descent have a larger value of n, Corollary 3 says that gradients via large-batch will have a better generalization than gradients via small-batch. Since the whole paper is centered around the theme - "good generalization ==> better performance", it does contradict Keskar et al.’s results.
> >
> > 3) Well, the paper initially didn't say anything about any validation set for choosing values of lambda. Can the code be shared on anonymously?
> >
> > 4) For true labels, the decrease in training accuracy (.021) and the decrease in generalizing error is (.007) are comparable. If 0.021 is not 'really compromising accuracy', then 0.007 is not really decreasing generalizing error.
> >
> > Also, for random labels the test accuracy will always be around .1 and one could keep decreasing the training accuracy (to 0.1) by increasing the value of lambda in l_2 norm (all the way upto infinity). Hence I don't see why one should use the norm proposed in this paper over l_2.
> >
> > 5) Your comment is unclear here.
> >
> > Minor points
> >
> > 1) Technically Gaussian is still not a band-limited function (by the definition stated in the paper). Restricting ourselves to Borel measurable functions for now, any function in L_2 can be arbitrarily 'approximated' by band-limited functions.
> >
> > 2,3) But still, what is the interpretation of such an (approximate) assumption?

---

> > > ### Author Response · Authors · 2017-11-24
> > > **Numerical experiments support theoretical results**
> > >
> > > “Simplifying ReLU by sinusoidal function doesn't seem like a good idea and seems very forced.”
> > >
> > > In fact, our Fourier-based approach suggests the opposite is true. Theorem 3 proves a path norm-based generalization bound for ReLU activation by considering the Fourier representation of ReLU function. Note that Fourier transform is a function’s representation in the sinusoidal basis.
> > >
> > > “one could keep decreasing the training accuracy (to 0.1) by increasing the value of lambda in l_2 norm”
> > >
> > > Yes, but for L2-norm if some lambda coefficient closes the generalization gap for random labels, that lambda will lead to a considerable drop in test accuracy for true labels. On the other hand, for path norms we numerically showed that the same lambda coefficient can close the generalization gap for both true and random labels without compromising test accuracy for true labels (Figures 1B and 3).
> > >
> > > “Corollary 3 says that gradients via large-batch will have a better generalization than gradients via small-batch”
> > >
> > > We emphasize that Corollary 3 only claims the generalization of the GRADIENT of the empirical risk to the GRADIENT of the population risk. This is a sufficient condition, which holds only for large-batch gradient descent, for applying Theorem 4 to guarantee generalization for the local minima found by large-batch gradient descent.
> > >
> > > “Can the code be shared on anonymously?”
> > >
> > > Our code is much longer than the 5000 character limit of openreview comments. We will make the code publicly available on Github after the anonymous review process is complete.
> > >
> > > “5) Your comment is unclear here.”
> > >
> > > Your original comment was “From Fig 2b the authors conclude that random labels will have a higher path-norm.” However, figure 2b shows Fourier L1-norm which is completely different from path norms.
> > >
> > > “Restricting ourselves to Borel measurable functions for now, any function in L_2 can be arbitrarily 'approximated' by band-limited functions”
> > >
> > > This statement is incorrect. For example, consider the density function of a uniform random variable over [-0.5,0.5]. While this function is in L_2, its Fourier transform (the sinc function) is not absolutely integrable. Hence the function cannot be arbitrarily well approximated by bandlimited functions, since the absolute integral of its Fourier transform over [B,infty) is infinite for any value B.

---

> > > > ### Public Comment · (anonymous) · 2017-11-24
> > > > **Points not addressed satisfactorily.**
> > > >
> > > > "Theorem 3 proves a path norm-based generalization bound for ReLU activation by considering the Fourier representation of ReLU function." - The theorem is restricted to class of NNs -  F_\phi_\lapha. This class  seems to have been defined by reverse engineering the proof. Any interpretations  on class F_\phi_\lapha?
> > > >
> > > > "Yes, but for L2-norm if some lambda coefficient closes the generalization gap for random labels, that lambda will lead to a considerable drop in test accuracy for true labels."
> > > > Agreed. However, changing the regularization only to improve the generalization of random labels at the cost of loosing accuracy (by 5% in the paper's experiments) on real labels seems unpractical.
> > > > Ideally one should compare the GE of NNs with path-norm vs l2 on random labels such that the NNs have same accuracy (or GE) on real labels.
> > > >
> > > > "Our code is much longer than the 5000 character limit of openreview comments"
> > > > Sure. But many authors share their code via anonymous github accounts.
> > > >
> > > >
> > > > "This is a sufficient condition" - I see. But since we empirically know that small batch gradient descent outperforms large-batch, this condition might not the right condition to look to prove the observation by Keskar et al's mentioned in the introduction.
> > > >
> > > >
> > > > “From Fig 2b the authors conclude that random labels will have a higher path-norm.”
> > > > You are right, it was a typo. I meant "Fourier L1-norm".  Thus my comment changes to - From Fig 2b the authors conclude that random labels will have a higher Fourier L1-norm.
> > > >
> > > > "Hence the function cannot be arbitrarily well approximated by bandlimited functions" - I meant approximation in L2 sense (not L1) which is commonly used in Fourier analysis since it has a physical meaning of energy (My conclusions are based on Parseval's theorem). I know that proofs in the appendix of the paper requires approximation in L1 sense but thats not at all clear in the introduction.

---

> > > > > ### Author Response · Authors · 2017-11-27
> > > > > **Numerical experiments support theoretical results**
> > > > >
> > > > > “This class seems to have been defined by reverse engineering the proof. Any interpretations on class F_\phi_\lapha?”
> > > > >
> > > > > F_\phi_\lapha is the set of 2-layer neural nets (ReLU-type activation) with bounded group path norm. That’s why in our earlier response we called the generalization result in Theorem 3 “a path norm-based generalization bound.”
> > > > >
> > > > > “at the cost of loosing accuracy (by 5% in the paper's experiments) on real labels”
> > > > >
> > > > > As we stated in our previous response, “due to computational constraints, we could only test a small set of lambda values for each regularization strategy.” By considering larger validation sets for lambda, we can tune lambda to get closer validation accuracy to the original accuracy.
> > > > >
> > > > > “But many authors share their code via anonymous github accounts.”
> > > > >
> > > > > We will check this option. Let us repeat that we will make the code publicly available on our Github account after the anonymous review process is complete.
> > > > >
> > > > > “From Fig 2b the authors conclude that random labels will have a higher Fourier L1-norm.”
> > > > >
> > > > > We have not concluded “random labels will have a higher Fourier L1-norm” from figure 2b. The only conclusions made from figure 2b in our manuscript (section 7.1) are “Figures 2b and 2c confirm that both Fourier L1-norm and bandwidth consistently increase with training” and “This suggests that, as implied by the theory above, regularizing Fourier L1-norm and bandwidth could improve generalizability of the final learned model.”
> > > > >
> > > > > “I know that proofs in the appendix of the paper requires approximation in L1 sense thats not at all clear in the introduction.”
> > > > >
> > > > > We will make this clearer in the text.

---

### Public Comment · (anonymous) · 2017-12-26
**Theoretical results and experiments aren't consistent (mostly contradicting).**

Following your previous comments  - "We will make the code publicly available on Github after the anonymous review process is complete."
Can you make the code public?

---

> ### Author Response · Authors · 2018-01-05
> **Review process continues till the end of January**
>
> The anonymous review process continues till the end of January. As we stated before, we will share the code on our Github account after the review process is complete.

---

### Author Response · Authors · 2018-01-05
**New revision posted**

We thank all the reviewers for their detailed comments and feedback which greatly helped us to improve this work. We have posted a revision which includes changes addressing the main comments. In summary,

1) We have clarified the explanation after Corollary 2 about the comparison between the Fourier-based Rademacher complexity bound and the existing Lipschitz-based bounds for 2-layer neural nets (asked by AnonReviewer2 and AnonReviewer3).

2) We have added Remark 1 after Theorem 4 explaining what this result implies when X has a multivariate Gaussian distribution (asked by AnonReviewer3).

3) We have made the explanation after Corollary 3 clearer (asked by AnonReviewer1) by directly applying an approximate version of Theorem 4 included in section 8.6 (asked by AnonReviewer3) to the local minima of the empirical risk.

---

### Decision · Program_Chairs · 2018-01-29
**ICLR 2018 Conference Acceptance Decision**

**Decision:**

Reject

**Comment:**

Understanding the generalization behavior of deep networks is certainly an open problem. While this paper appears to develop some interesting new Fourier-based methods in this direction, the analysis in its current form is currently too restrictive, with somewhat limited empirical support, to broadly appeal to the ICLR community. Please see the reviews for more details.